

# The complex origin and spatial distribution of non-pure sulfate
# particles (NSPs) in the stratosphere

Jean-Baptiste Renard[1], Gwenaël Berthet[1], Anny-Chantal Levasseur-Regourd[2], Sergey
Beresnev[3], Alain Miffre[4], Patrick Rairoux[4], Damien Vignelles[1,5], Fabrice Jégou[1]
1. LPC2E-CNRS, Orléans, France
2. Sorbonne Université, LATMOS-IPSL, Campus Pierre et Marie Curie, Paris, France
3. Ural Federal University, Institute of Natural Sciences and Mathematics, Yekaterinburg,
Russia
4. University of Lyon, Université Claude Bernard Lyon 1, CNRS, Institut Lumière Matière,
Villeurbanne, France
5. MeteoModem Company, Ury, France
## Abstract
While droplets with pure mixtures of water and sulfuric acid are the main component
of stratospheric aerosols, field measurements performed for more than 30 years have
shown that non-sulfate materials, thereafter referred to by us as NSP (for "Non-pure Sulfate
Particles", not considering frozen material) are also present. Such materials, which are
released from both the Earth through volcanic eruptions, pollution or biomass burning, and
from space through interplanetary dust and micrometeoroids, present a wide variety of
composition and shape, with sizes ranging from several nm to several hundreds of μm. No
single instrumental technique, from ground, from airplanes, under balloons and onboard
satellites using remote-sensing and in-situ instruments. can provide alone a global view of
the stratospheric NSPs, which exhibit a strong variability in terms of spatial distribution and
composition. To better understand the origin of the NSPs, we have performed new field
measurements from mid- 2013 with the Light Optical Aerosol Counter (LOAC) instrument
during 135 flights carried out under weather balloons at various latitudes and up to altitudes
of 35 km. Coupled with previous measurements obtained with the Tropospheric and
Stratospheric Aerosols Counter (STAC) under stratospheric balloons in the 2004 - 2011
period, the LOAC measurements show the presence of stratospheric layers presenting
enhanced-concentrations associated with NSPs, with a bimodal vertical repartition centered
by 17 and 30 km altitude. Also, large particles are detected, with sizes up to several tens of
μm, with decreasing concentrations with increasing altitudes. Such observations, which are
not correlated with meteor shower events, could be due to dynamical and photophoretic
effects lifting and sustaining particles mainly coming from the Earth. When combining all the
detections in the stratosphere from different methods of measurements, we may conclude
that the concentrations and the vertical distributions of NSPs are highly variable and do not
match the estimated concentrations of material in space at Earth orbit. The paper ends by
highlighting some open questions on these stratospheric materials and presents some
possible new strategies for frequent measurements, to confirm that NSPs are indeed mainly
of terrestrial origin, and to better circumvent the NSPs impact on stratospheric chemistry
and on the Earth's climate.



## 1. Introduction

The Earth's stratospheric aerosols mainly contain liquid particles as pure mixture of water and sulfuric acid particles (hereafter referred to as sulfate aerosols or sulfate particles). They largely originate from natural and anthropogenic emissions of oxidized carbonyl sulfide (e.g. Kremser et al., 2016; Günther et al., 2018) or are released from volcanic eruptions injecting the $SO_2$ aerosol precursor directly into the stratosphere (Russell et al., 1996; Deshler et al., 2003; Solomon et al., 2011; Jégou et al., 2013; Bègue et al., 2017), with an altitude of injection controlled by the eruption intensity. Aerosols from volcanic eruptions of moderate amplitude (i.e. about 20 times less $SO_2$ injected than for the Pinatubo major volcanic eruption in 1991) occurring regularly since the year 2000, modulate the sulfate aerosol concentration at the global or hemispherical scale over periods of months (Vernier et al., 2011). Nevertheless, observations of the aerosol content show that particles concentration remains presently low in the northern hemisphere stratosphere, as compared to the post-1991 period impacted by the Pinatubo volcanic eruption. Hence, in what follows, the current stratospheric conditions are typically referred to as "background" aerosol conditions, with sulfate particle concentrations expected to decrease with increasing altitude above 20 km (Deshler et al., 2006).

Although sulfate particles are the main component of the stratospheric aerosols, at least in the lower and middle stratosphere, remote sensing and in-situ measurements performed for more than 30 years have shown that materials (not considering frozen material) clearly stand out from the sulfate population in terms of composition and optical properties, with abundances depending on the altitude, latitude, and season (Baumgardner et al., 2004; Curtius et al., 2005; Murphy et al., 2007; Renard et al., 2008; Ebert et al., 2016; Schütze et al., 2017). Such materials, that we define here as NSPs, acronym for "Non-pure Sulfate Particles", could be externally-mixed (i.e. pure solid and liquid particles) or internally-mixed (i.e. included in or coated by pure sulfate). This definition clusters the various complex properties, mainly in terms of composition, shape and conditions of sublimation, reported in the literature. NSPs can be in the form of semi-volatile particles like some secondary organic aerosols, or volatile liquid particles formed from gaseous precursors, with or without dissolved material. They can contain refractory or non-refractory material in the form of amorphous, compact, aggregated or fractal solid parts like black carbon, thereafter BC, or soot particles (black carbon refer to pure carbon particles, while soot are carbonaceous particles including carbon and other materials). These particles are mostly optically-absorbing conversely to pure sulfate particles.

NSPs can have different origins, coming either from the Earth or from space. As detailed below, the strong variability in the corresponding concentration and chemical composition measurements could be due to the various sources, including possible very localized contributions. The expected low concentrations of these particles, the wide variety of their shape and their chemical compositions and the various sensitivity of the measurement techniques have not, up to now, made it possible to reach a comprehensive vision of non-sulfate materials in the stratosphere in terms of origins, content, physical properties, composition and seasonal/inter-annual variability, which would be necessary to address different questions such as radiative effects.

The aim of this paper is to contribute to a better understanding of NSPs origins and variability in the stratosphere. For that purpose, we have developed a new strategy of measurements, using the Light Optical Aerosols Counter (LOAC), which has performed 135





flights under weather balloons over the mid-2013 – mid-2019 period. We first present the methodologies already used for the NSPs detection; then, we present the LOAC instrument and the results obtained for concentrations, size distribution, temporal and spatial variabilities; finally we discuss the possible origins of such variabilities, and the contribution of LOAC to better estimate the source of the NSPs.

## 2. Present methodologies for NSPs detection in the stratosphere

### 2.1 Context

NSPs have been detected using different remote-sensing and in-situ instruments, from the ground, onboard airplanes, under balloons and onboard satellites. None of these instruments can solely characterize these particles at all possible scales, from the detection of specific events to the observation of the spatial and temporal variability of their concentrations, size distributions and chemical composition. Then, all these measurements must be combined to tentatively propose a comprehensive view of NSPs in the stratosphere.

### 2.2 Ground-based measurements

Ground-based observations provide sparse or time-series measurements from a given place, meaning that fortuitous events of stratospheric aerosol enhancements with or without NSPs can be detected.

Remote-sensing photometric measurements at twilight can provide vertical profiles of twilight intensity that reflect the aerosol abundances. However, such measurements have been seldom conducted, mainly during meteor shower episodes, showing strong and transient aerosols enhancements in the middle and upper stratosphere (Padma Kumari et al., 2005, 2008).

Other transient episodes have been reported from lidar measurements, in relation with the disintegration of a meteorite (Klekociuk et al. 2005), or the debris of rockets or satellites (Gerding et al., 2003), or wildfire plume events (Siebert et al., 2000; Khaykin et al., 2018; Haarig et al., 2018). Most of these measurements are performed in the framework of the Network for the Detection of Atmospheric Composition Change (NDACC) or European Aerosol Research Lidar Network (5EARLINET). There is however no direct determination of the precise size distribution and concentration of the particles, although indications on the altitude dependency of their nature (liquid, solid, mixed) and their mean size values can be derived from the depolarization they induce on the backscattered laser light (Stein et al., 1994). Such analyses are conducted from an optical point of view by partitioning the stratospheric aerosol into icy particles and spherical particles containing sulfate component, which backscatter light is well-known, and non-sulfate optical component that mainly refers to light-absorbing material including BC and soot. In particular, such a partitioning was already performed by Miffre et al. (2015) in the troposphere when coupling incandescence and lidar field measurements.





### 2.3 Airplane measurements

Airplanes can perform long horizontal excursions but shorter vertical profiles excursions in the lower stratosphere up to about 22km altitude, by carrying instruments for in-situ measurements. The measurements are conducted during dedicated field campaigns, thus providing accurate but sparse measurements that might not be representative of the whole lower stratosphere. The main difficulty could be the collection of very large particles greater than several tens of µm, although specific collecting methods have been developed to limit the possible breaking of the particles due to relative speeds of up to 200 m/s (e. g. Scott and Chittenden, 2002).

Historically, the first campaigns were conducted to collect interplanetary dust, with subsequent laboratory analysis by electronic microscopy (TEM/SEM) and energy-dispersive X-ray microanalysis (Brownlee, 1985; Warren and Zolensky, 1996). NASA has been collecting dust in the stratosphere since the beginning of 1981, with U-2, ER-2 and WB-57 airplanes. These flights have mostly ranged over most of the USA (as far as north as Alaska) and Central America. The main challenging tasks are to distinguish between the refractory material coming either from space or from the Earth itself and to determine the natural or anthropogenic origin of these particles through morphology and composition analyzes (Pueschel et al., 1992; Blake and Kato, 1995; Pueschel et al., 1997; Strawa et al., 1999; Ebert et al., 2016; Schütze et al., 2017).

In-situ optical counting instruments onboard airplanes provide the size distribution of the particles in the lower stratosphere, for particles greater than about 0.2 µm. Conventional counters are highly sensitive to the complex refractive index of the particles; BC particles can typically be up to 10 times darker than liquid sulfuric-acid aerosols of the same size. To accurately retrieve the size distribution, the nature of the detected aerosols must be known when processing the data, otherwise the contribution of the optically-absorbing particles could be strongly underestimated. A more sophisticated methodology consists in using a first channel specifically for the determination of the total aerosol concentration and a second one heated to 250°C detecting the presence of non-volatile materials. The fraction of stratospheric particles not composed entirely of volatile (i.e. water and sulfuric acid) material is then estimated by calculating the difference between both channels (Curtius et al., 2005).

The in-situ scattering and incandescence techniques, where the aerosols are also heated, is used to determine the size distribution of the submicron particles down to 0.2 µm, the proportion of light-absorbing refractory NSPs, the determination of bulk composition (such a BC) and information about the possible presence of coating  (Baumgardner et al., 2004; Schwarz et al., 2006; Weigel et al., 2014).

Finally, the in-situ mass spectrometry provides the composition of the particles and the vertical profile of the partitioning between various families of NSPs containing or not carbonaceous material and metals (Murphy et al., 1998; Jost et al., 2004; Murphy et al., 2007; Murphy et al., 2014). However, from this technique it is difficult to conclude whether the metallic elements are dissolved or are in the form of refractory inclusions in the sulfate particles.


### 2.4 Balloon-borne measurements

Stratospheric (zero pressure) balloons can reach 40 km altitude and stay up to a few tens of hours in flight. They are well-adapted to study the middle stratosphere above altitudes reached by airplanes. Measurements can be conducted during ascent, at float and during a slow descent. However, technical and operational constraints strongly restrain the number of flights and the geographic zones for launches.

Remote sensing measurements, using natural light sources (Sun, Moon, stars) can provide the vertical profile of aerosols extinction with a resolution of a few hundreds of meters, generally in the UV-visible domain (Renard et al., 2002). For liquid aerosols, the size distribution can be retrieved using Mie scattering calculations. Such an observational method needs to assume that the stratosphere is composed of horizontally homogeneous layers along lines of sight of tens to hundreds of km long, although significant local concentration variations can bias the retrieval (Berthet et al. 2007). Nevertheless, non-monotonous UV-visible extinctions could be an indicator of the possible presence of optically absorbing material in the stratosphere together with the sulfate aerosol population (Berthet et al., 2002; Renard et al., 2002). Also, the measurements of the local radiance scattering function for the aerosols (Mishchenko, et al., 2004) can be used to distinguish between sulfate and other types of particles (Renard et al. 2008). However, such studies have not addressed the possibility for the aerosols to be internally or externally-mixed.

The optical aerosol counters are easier to use from balloons than from airplanes, since the relative speed between the instrument and the ambient air is low, around 5m/s during ascent or slow descent and close to zero at float. They can typically detect particles with sizes from about 0.2 μm to a few μm (Deshler et al., 2006; Renard et al., 2005), and provide the vertical profiles of the aerosols for several size classes. An improved optical particle counter has been used to detect the fraction of aerosols that are charged, probably by the galactic cosmic rays (Renard et al., 2013). Such charged particles could have some implication in the high-energy phenomena in the middle and upper atmosphere (Füllekrug et al., 2013).

Finally, the negligible speed between the balloon and the ambient air at float altitude is optimal to collect the particles without breaking them. As for airplane collection, the particles are analyzed in the laboratory after a soft landing of the gondola (Testa et al., 1990; Ciucci et al., 2008; Della Corte et al., 2013).

### 2.5 Satellite measurements

Satellite instruments can provide a global coverage of the aerosol content in the stratosphere (Bingen et al., 2004; Vanhellemont et al., 2010; Vernier et al., 2011; Salazar et al., 2013; Thomason et al., 2018). They can be used to derive trends over several years or to study locally strong sources of aerosols. For such remote sensing measurements, inversion methods are necessary to retrieve the vertical profiles, using assumptions on the complex refractive index of the particles and/or on the shape of the size distribution (e.g. Bourassa et al., 2012). Nevertheless, they cannot access to the local variability of the aerosol content, which can be potentially diluted along the line of sight and/or could be removed when applying smoothing or filtering procedures.





The UV-visible extinction measurements obtained from occultation or from limb
profiling rely on the hypothesis of homogeneous layers in the stratosphere. The vertical
resolution is between one and a few km. Nevertheless, it is possible to follow intense events
of injection of refractory material in the stratosphere from fires (carbonaceous particles),
volcanoes (ash) and meteoroid disintegration (Fromm and Servranckx, 2003; Fromm et al.
2006; Niemeier et al., 2009; Gorkavyi et al., 2013; Rieger et al., 2014). Also, extinction
measurements can be used to search for the presence of NSPs with respect to the pure
sulfate population in the middle stratosphere (Neely et al., 2011).
The space-borne lidar measurements, like the CALIOP instrument onboard the
Calipso satellite, are mainly dedicated to cloud studies, tropospheric aerosols and the
boundary layer (e. g. Bourgeois et al., 2018), since the scattered signal is often too low for
the detection of stratospheric aerosols. Also, reference altitudes used to derive the Rayleigh
signature in the lidar retrieval and assumed to be aerosol-free are often too low to detect
stratospheric aerosols in general above 30 km altitude (Vernier et al., 2009). Nevertheless,
analyses can be conducted during specific events such as a volcanic eruption and injections
of carbonaceous particles and/or the gaseous precursors by the Asian monsoon or pyro-
convection (Vernier et al., 2016; Govardhan et al., 2017; Vernier et al., 2018; Khaykin et al.,
252 2018).

Finally, attempts to collect dust from space entering the Earth's atmosphere were
made from Gemini 10, Skylab, Salyut 7 and MIR space stations, and from the retrieval of
space exposed surfaces of satellites. The Long Duration Exposure Facility (LDEF) was exposed
for almost 6 years at altitudes ranging from 580 km to 332 km, and provided evidence for
micrometeoroids (Mandeville et al., 1991; Love and Brownlee, 1993; Kalashnikova et al.,
2000) that do enter the Earth's atmosphere.

## 3. Regular flights with balloon-borne aerosols counters

### 3.1 Instruments for stratospheric studies

The previous measurements have shown that the stratospheric content of NSPs
exhibits a strong horizontal, vertical and temporal variability, since the sources of NSPs can
be diverse. Regular and frequent in situ measurements are of high relevance to derive the
stratospheric aerosols content and to follow its evolution, because the use of a priori
hypothesis on the shape, the composition and the size distribution of the particles
commonly used in data retrievals from remote sensing instruments (e.g. Bourassa et al.,
2012) is at least minimized or at best pointless. Such measurements strategy with aerosols
counters under balloons started about 50 years ago.
The University of Wyoming aerosols counters (Deshler et al., 2003; Deshler et al.,
2006) and the Stratospheric and Tropospheric Aerosols Counter (STAC) (Renard et al., 2008;
Renard et al., 2010) have provided locally the size distribution and the concentrations of
aerosols up to 40 km in altitude when launched under (large) stratospheric balloons. In
particular, during its 21 flights in the 2004 - 2011 period, STAC has often detected strong
aerosol concentration enhancements over a vertical extent from few hundreds of meters to
few km.
Such instruments were calibrated for the detection of liquid particles typically in the
0.2 − 5 µm size range. Since their measurement technique is sensitive to the complex





refractive index of the particles, these instruments cannot be used to distinguish between
transparent liquid particles and optically absorbing NSPs, the size of the latest being possibly
underestimated. The weight of the instrument is of several kg, preventing them to be used
under small balloons as weather balloons.
Given that the aerosol variability does not necessarily vary homogeneously
throughout the stratosphere especially under the influence of sporadic events (volcanic
eruptions, fires, meteoroid disintegrations), regular and frequent in-situ measurements must
be conducted from several locations in the world. Accounting for operational constraints and
cost issues, it seems that the most valuable method is to launch light and inexpensive
instruments under weather balloons, sending the data by a telemetry system, with the risk
of losing the instrument after the flight. Such types of balloons can reach an altitude of 35
km, but the burst of the balloon cannot be controlled. The payload weight must be below a
few kg to account for the international aeronautic rules. The cost for the balloons and the
gas are no more than a few hundreds of euros. Two or three people are necessary to launch
the balloon from almost everywhere in the world. The wind speed at ground could be up to
15 m/s. Then, frequent measurements, typically tens of flights per year, can be conducted
regularly and during specific events almost from everywhere.
Light instruments are not currently available for collecting systems or for mass-
spectrometry. On the opposite, two light optical aerosols counter with a weight of about 1
kg are now available on the market. The first one is the Printed Optical Particle Spectrometer
(POPS), designed for the detection of liquid particles in the ~0.15 − 1 μm size range (Gao et
al., 2016). It can provide very accurate size distributions and concentrations of sub-micronic
sulfate stratospheric aerosols with a vertical resolution of 100 m or better, but is not
designed to detect the largest particles previously detected in the stratosphere (e.g.
Jessberger et al., 2001; Ciucci et al., 2008) and cannot identify the optically absorbing
particles. The second one is the Light Optical Aerosols Counter (LOAC). LOAC is a novel
instrumental concept (Renard et al., 2016a), providing the size distributions, the
concentrations, and an estimate of the typology, for particles in the 0.2 - 50 μm size range.
LOAC uses a statistical approach to retrieve the concentration of particles smaller than 1 μm
(Renard et al., 2016a). When the concentration of submicronic particles is low, typically
below 10 particles cm$^{-3}$ for sizes greater than 0.2 μm, the integration time must be increased
up to 10 min; then the vertical resolution is between 1 km and 3 km for a balloon ascent
speed of about 5 m/s. Since LOAC is not sensitive to the complex refractive index of the
particles, it can detect all particle types. LOAC is well appropriated for the detection of the
NSPs, as previously shown for dust particles in the troposphere (Renard et al., 2018), while
the POPS is better designed for the detection of the submicronic sulfate aerosols; then these
two instruments could be considered as complementary.
**3.2 The LOAC instrument**
The particles are injected through an optical chamber by a pumping system, cross a
laser beam and the light scattered by the particles is recorded by two detectors.
Conventional aerosols counters typically performed measurements at large scattering angles
(greater than 30° and often around 90°). Since the scattered light is sensitive to the size of
the particles but also to their complex refractive index and their shape including porosity
effects, conventional optical counter measurements must be corrected for the nature of the





particles. On the opposite, LOAC performs measurement at small scattering angles, in the
11°-16° range, where the scattered light is mainly coming from diffraction that does not
depend on the complex refractive index nor on the porosity of the irregular-shaped particles
(Lurton et al. 2014). As a result, a direct correspondence between the intensity of the
scattered light and the optical diameter of the particles becomes feasible. LOAC provides
particles number concentrations for 19 sizes in the 0.2 – 50 μm size range, with an
uncertainty of ±20% for concentrations higher than 10 particles $cm^{-3}$; the uncertainty
increases to about ±30% for submicronic particle concentrations higher than 1 particle $cm^{-3}$,
and to about ±60% for concentrations smaller than $10^{-2}$ particle $cm^{-3}$. The size of the
particles provided by LOAC is an optical diameter, which could differ from aerodynamical,
electric mobility and gyration diameters used by other counting techniques in case of
irregular particles. Also, the refractory particles could be hydrated, thus having a size greater
than dry particles. The ability of LOAC to accurately detect micron-sized particles and larger
particles has been validated during numerous intercomparaison sessions with different
instruments (Renard et al., 2016a; Renard et al., 2018).

LOAC has a second detector at a scattering angle in the 50°-70° range, where the
scattered light is very sensitive to the complex refractive index and to the porosity of the
particles. By statistically combining these measurements with those at 11-16°, we obtain a
parameter called "speciation index", which is representative of the properties of the
particles to absorb light (Renard et al., 2016a). Higher is the speciation index, darker are the
particles. Speciation index reference measurements were conducted in laboratory with pure
reference samples to establish a data base. By comparing the ambient air measurements to
the database, we can tentatively identify the basic nature of the particles, or typology. As
confirmed during tests in laboratory and in ambient air, LOAC can indicate if the detected
aerosols are icy, are in a non-optically absorbing liquid phase as the typical stratospheric
pure sulfate population, are semi-transparent NSPs as some dry minerals or highly hydrated
solid aerosols, or are optically-absorbing NSPs as carbonaceous particles. For the last case, it
is not possible however to know if the particles are externally or internally-mixed with
sulfate, or can be considered as a secondary organic aerosol. This approach is just a first step
to validate the ability of optical measurements to provide an estimate of the nature of the
stratospheric aerosols.

The raw LOAC concentrations are corrected of the sampling efficiency when the
measurements are conducted under weather balloon (Renard et al., 2016a), the sampling
being dominated by sub-isokinetic conditions and the divergence of the flow field at the inlet
entrance.
**3.3 Vertical profile of aerosol concentrations obtained with LOAC**
The LOAC gondola includes batteries, telemetry to send the data in real time, and
temperature and humidity sensors, using the MeteoModem Company system (Renard et al.,
2016b). Since May 2013 to mid-2019, 135 flights reaching the stratosphere have been
successfully conducted from France, from Spain and from Ile de la Réunion (Indian Ocean).
Regular flights, from one to four per month, have been operated since February 2014 from
France mainly by Centre National d'Etudes Spatiales (CNES), the French Space Agency, from
its balloons launching base at Aire sur l'Adour (43.70°N, 0.25°W); in this case the balloons
are called "Light Dilatable Balloons" since they carry a scientific instrument, to distinguish





them for conventional weather balloons. Figure 1 presents a LOAC launch from Aire sur
l'Adour on 6 February 2014.

Figure 2 presents an example of the vertical evolution of the particle number
concentrations for a LOAC flight in 17 August 2017, again from Aire sur l'Adour.
Concentrations decrease with altitude, as expected for sulfate aerosols. The retrieved
typologies in the stratosphere indicate that the submicron aerosols are indeed transparent
liquid droplets based on the comparison with reference curves obtained in the laboratory
with this type of particles (Figure 3). Note that in this example the number of detected
particles for size classes above 3 μm is too low for the typology determination.

Few particles greater than 5 μm are detected above the tropopause, and one particle
larger than 40 μm is present at 25 km altitude. Since the flight was conducted while the
permanent Perseids meteor shower took place, one could suggest that LOAC has detected
some dust particles coming from space.

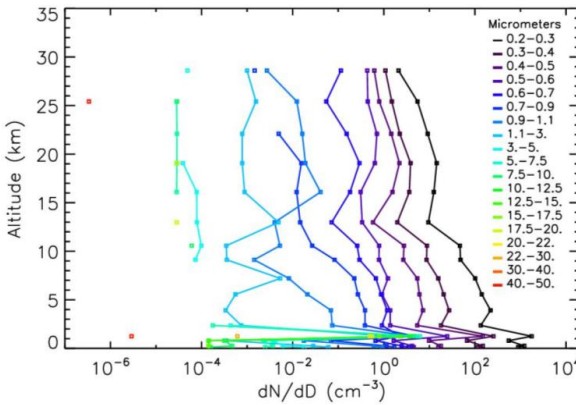

*Figure 1: LOAC launch from Aire sur l'Adour (France; 43.70°N, 0.25°W) on 6 February 2014*

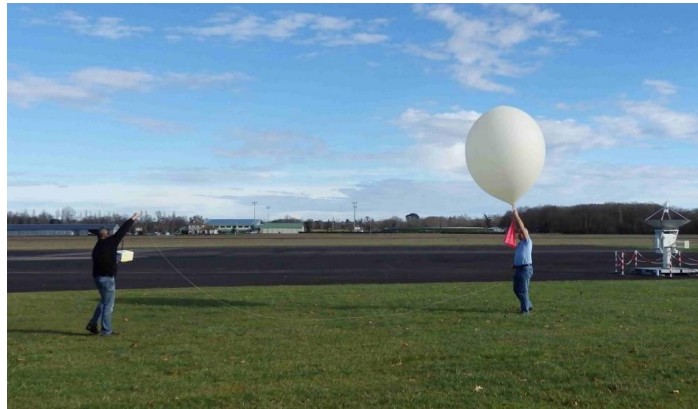

*Figure 2: Vertical profile for the LOAC 17 August 2017 flight from Aire sur l'Adour (France;*
*43.70°N, 0.25°W) during the Perseids meteor shower period. Errors bars (see text) are*
*omitted for clarity reasons*





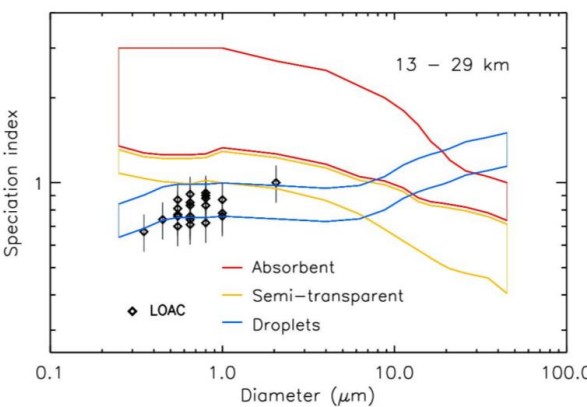


*Figure 3: Indications of typology from the LOAC speciation index in the stratosphere for the 17 August 2017 flight*


LOAC has detected some strong vertical variability of aerosol concentrations during
flights in background conditions (without recent volcanic eruptions). We consider here that a
strong concentration enhancement is detected whenever the concentrations are at least 5
times higher than the background concentrations measured during the same flight for at
least 5 consecutive size classes (this criterion is to ensure that the enhancements are real
and are not due to noise measurement fluctuations). We exclude the measurements
conducted at the edge of the polar vortex where the local dynamical variability can affect
the aerosols content (Renard et al., 2008).

Figure 4 presents an example of a strong concentration enhancement in the lower
stratosphere at an altitude of 18 km, as observed during a flight conducted from Aire sur
l'Adour on 11 August 2016, during the Perseids period. Several particles bigger than 5 $\mu$m
have been also detected at this altitude, which could result from the fragmentation of a
larger fluffy particle or could be an accumulation layer of carbonaceous particles (note that
the large particles between 8 and 12 km altitude correspond to a cirrus cloud). The typology
(Figure 5) indicates that particles up to 2 $\mu$m are indeed in liquid phase, in this case certainly
sulfates, while biggest ones are classified as strongly optically-absorbing NSPs.

Figure 6 presents another example of concentration enhancement observed in the
middle stratosphere at an altitude of 28 km and only for submicronic particles, during a flight
on 23 November 2017 from Aire sur l'Adour. This time, the typology (Figure 7) indicates
mainly optically-absorbing particles for the smaller size classes.

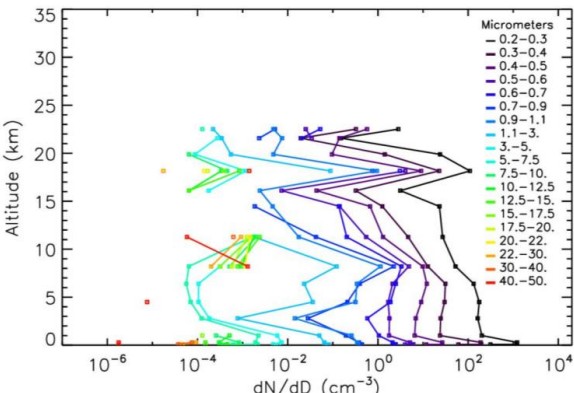

*Figure 4: Vertical profile for the LOAC 11 August 2016 flight from Aire sur l'Adour (France;*
*43.70°N, 0.25°W) during the Perseids meteor shower period. Errors bars (see text) are*
*omitted for clarity reasons*

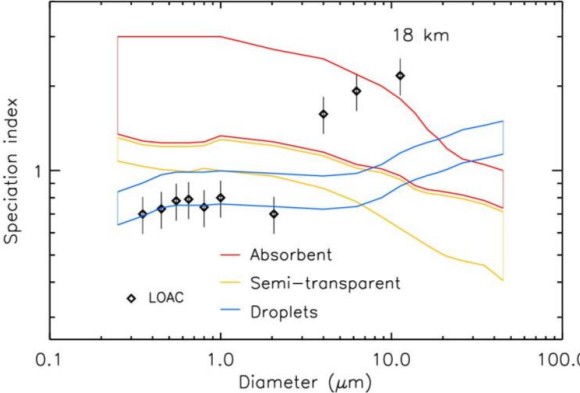

*Figure 5: Indications of typologies from the LOAC speciation index in the stratosphere for the*
*11 August 2016 flight*

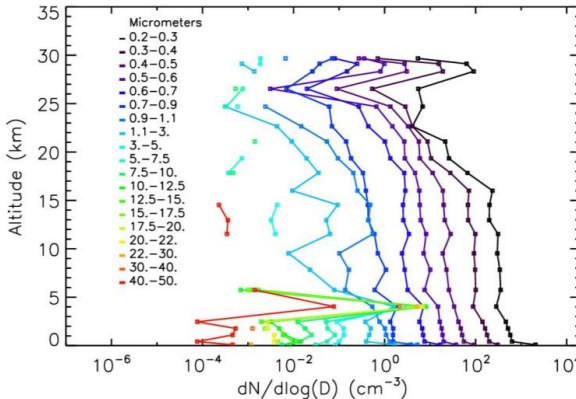

*Figure 6: Vertical profile for the LOAC 23 November 2017 flight from Aire sur l'Adour (France;*
*43.70°N, 0.25°W). Errors bars (see text) are omitted for clarity reasons*






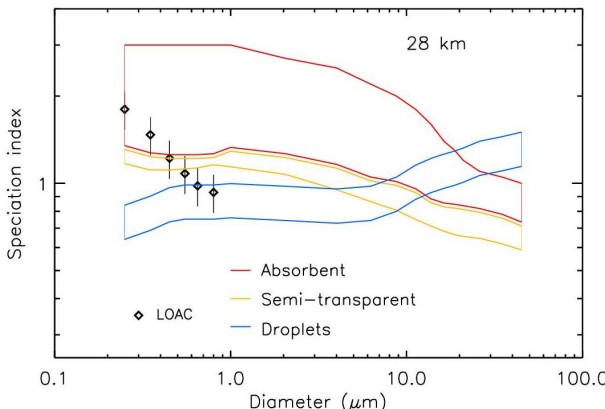


*Figure 7: Indications of typologies from the LOAC speciation index in the stratosphere for the 23 November 2017 flight*

As a result, within the error bars, we can conclude that LOAC seems to have detected similar concentration enhancements as those previously reported from STAC measurements, although the two instruments use different geometries of observation. About 25% of the 135 LOAC and 21 STAC flights exhibit such strong enhancements. The enhancements could be sometimes only for submicronic particles, and sometimes for all size classes up to more than 10 μm. The typology measurements indicate that most of the enhancements are dominated by NSPs particles: nevertheless, the speciation index varies from one flight to another from semi-transparent to strongly absorbent particles, which could indicate that several families of NSPs were detected. To attribute the origin of these detections, it is necessary to review the already published results on NSPs detections obtained by the various instrumental techniques.

## 4. Previous studies on the spatial and temporal variability, size and composition of NSPs

### 4.1 Vertical dependence of NSP particles

In the middle and upper stratosphere, Neely et al. (2011) have compared the SAGE II satellite extinctions to model calculation (WACCM) assuming only liquid aerosols. They have found strong enhancement of measured extinction above 30 km altitude on average; a mean extinction value of about $5.10^{-6}$ km$^{-1}$ is found at 40 km altitude although model calculations provide a zero value for sulfate aerosols. The extinction enhancements are associated with the contribution of meteoritic smoke particles, coming from the disintegration of meteorites or micrometeorites and recondensation processes (Plane, 2003; Plane et al., 2018; Rapp et al., 2007; Bardeen et al., 2008; Plane, 2012), while an external and/or internal mixture of smoke particles and sulfate aerosols occur at lower altitudes. The ubiquitous presence of non-optically transparent particles was also found in the extinction measurement by





GOMOS/Envisat, at the same altitudes as SAGE II, for which the wavelength dependence of
the extinction strongly differed from the one expected for liquid aerosols only (Salazar et al.,
2013). The presence of light-absorbing particles in the middle stratosphere was also
detected from sparse balloon measurements with extinctions and radiance measurements
(Renard et al., 2005; Renard et al., 2008); these particles could be either internally or
externally-mixed with the sulfate aerosols. Nevertheless, such remote sensing
measurements do not provide the precise nature and composition of these particles, so-
called "soot" by the authors although "strongly optically-absorbing material" could be more
appropriate (the terminology for describing such light-absorbing carbons is reviewed in Bond
and Bergstrom (2007)).
The presence of soot and BC in the lower stratosphere was detected below 20 km
altitude using wire impactors on airplane (Pueschel et al., 1992; Blake and Kato, 1995;
Strawa et al., 1999). According to Strawa et al. (1999), BC aerosol number density could be
of about 1 % of the total aerosols content in the lower stratosphere. On the other hand,
Baumgardner et al. (2004) have studied the concentration of light-absorbing particles
(attributed to BC and particles with metals) in the 0.2-0.8 μm size range above the
tropopause in the northern polar vortex by light scattering and incandescence
measurements, and found more than 10 particles cm$^{-3}$ in that size range. These light
absorbing particles are more concentrated by a factor of 10 than the non-light absorbing
particles below 0.3 μm inside the polar vortex, with higher contents than for extra-vortex air.
Similarly, by using counting techniques and heating to remove the volatile material, Curtius
et al. (2005) found a much higher fraction of particles containing non-volatile residues inside
than outside the polar vortex. These particles could result from the downward transport of
refractory meteoritic material within the polar vortex from the mesosphere to the lower
stratosphere as concluded from aerosol collections (Weigel et al., 2014; Ebert et al., 2016).
At other latitudes, Murphy et al. (2007, 2014) have derived two categories of NSPs
for altitudes below 20 km at different latitudes, using Particle Analysis by Laser Mass
Spectrometry (PALMS) observations: NSP with metal compounds proposed to probably
originate from vaporized and condensed meteoritic material, and NSP with mixture of
sulfate and organic particles. The second category corresponds to the main components of
aerosols in the mid-latitude and tropical lower stratosphere. These observations agree with
the Schwarz et al. (2006) results in the northern tropical region, where 40% of BC particles
showed evidence of internal mixing. Then, the average fraction of carbonaceous material in
the stratospheric particles decreases rapidly with increasing altitude.
**4.2 Sporadic strong enhancements in the stratospheric aerosol content from NSPs**
Several authors have reported local enhancements of NSP concentrations in the
stratosphere. These sporadic features are highly variable in term of residence times (i.e.
from the scale of days to months).
Jost et al. (2004) have detected plumes of carbonaceous particles originating from
North American forest burning in July 2002 up to an altitude of 16 km by Laser Mass
spectrometry and counting measurement. The increase concentration is about 7 times
higher than background conditions. The origin of the particles was confirmed with correlated
CO measurements. Short-living and local increases up to a factor 2.5 in aerosol extinction
measurements related to intense biomass burning have been seen by the SAGE III space-



borne instrument over Australia at the beginning of 2003 (Fromm et al. 2006). These particles can be injected in the lower stratosphere by the pyroconvection process occurring at the top of the dense smoke clouds (Fromm and Servranckx, 2003). Ground-based lidar measurements at Observatoire de Haute Provence, France (43.9°N, 5.7°E) and space-borne lidar measurements from the CALIOP/Calipso instrument have detected plumes of fire particles between 18 and 20 km altitude over southern France, coming from wildfires in northwest Canada and United States in August 2017 (Khaykin et al., 2018). The scattering ratio at 532 nm wavelength is about 10 times higher than for background conditions. Also, a layer of soot particles at 15-16 km altitude coming from these wildfires, and well identified by the specific wavelength dependence of the lidar depolarization ratios, were observed in Germany over Leipzig (51.3°N, 12.4°) on 22 August 2017 (Haarig et al., 2018). These measurements have shown that local enhancements of fire plume particles can be detected several thousands of km from their sources, impacting the stratospheric aerosol content at the hemispheric scale, with an amplitude (both in terms of aerosol content and residence time) comparable to that of a moderate volcanic eruption (Peterson et al., 2018). Satellite data show that the NSPs from this specific fire plume event remained detectable in the northern hemisphere stratosphere over a period of about 8 months (Kloss et al., 2019). Nevertheless, the morphology and composition of these particles have not been determined so far.

Local intrusions of particles attributed to BC have also been reported from CALIOP/Calipso in the lower stratosphere during the monsoon season over India, with few sparse enhancements by 20-30 km altitude (Govardhan et al., 2017). The origin of such particles is not well-established though these authors propose them to originate from airplanes traffic before being vertically transported.

Moderate volcanic eruptions can inject aerosols directly into the lower stratosphere. Some ashes can be present, as observed after the Kelud eruption (February 2014) in the lower tropical stratosphere (Vernier et al., 2016). Ashes were detected by analysis of CALIOP/Calipso space-borne observation in comparison with in situ measurements from optical backscatter aerosols sounders and optical aerosols counters. The residence time of ash material in the stratosphere depends on the injection altitude and on the size of the particles. Values of several weeks are expected due to sedimentation (Niemeier et al., 2009). However, the residence time of ash material is not clearly determined especially when mixing or coating processes with sulfate occur.

Rockets produce solid particles that can be found in the stratosphere. Campaigns of in-situ particle counter measurements were conducted on board airplanes in 1996 and 1997 to detect the alumina particles in the motor exhaust plumes. Measurements in the stratosphere between 17 and 19.5 km showed strong concentration enhancements of about a factor of 100 with respect to the nearby background conditions (Ross et al., 1999).

The ground-based twilight photometric observations have shown accumulation layers at altitudes of 30 km and 54 km on 20-21 November 1998 during the Leonids meteor shower (Mateshvili et al., 1999). Large and thin dust accumulation layers were also detected between 20 and 50 km altitude on 21-26 November 2001, 25-27 November 2002, and 16-17 November 2003 during the Leonids (Padma Kumari et al., 2005). The Leonids, as opposed to permanent meteor showers visible on every year at a given epoch, may produce periodic meteor storms, for about 3 to 4 years, every 33 years, in November. The enhancements in the twilight light intensities are of tens of percent. Such layers were observed 4 to 8 days after the peak meteor activity, but with a strong variability from one day to another for the





altitude and the amplitude of the vertical structures. The authors state that, a couple of weeks after the meteor activity, the atmosphere had recovered its normal dust distribution profile although a dust layer at 30 km altitude could have persisted. Such observations of transient layers need to be confirmed for other occasional and major meteor storms.

Some lidar observations in the Arctic during the 2000-2001 winter have fortuitously detected strong concentration enhancements from 25 to 40 km altitude, with strong spatial and temporal variability over a few days (Gerding et al., 2003). It has been proposed that they could originate from meteoritic debris after the disintegration of a meteoroid in the atmosphere or from debris of condensed rocket fuel. Klekociuk et al. (2005) have also observed a strong particle enhancement around 30 km altitude, which was well identified as coming from the disintegration of a large meteoroid of a few meters in size on 3 September 2004 over Antarctica, with residence time from weeks to months.

Major meteoritic disintegrations can produce strong enhancements in aerosol concentrations, initially localized at the altitude of the disintegration and then progressively dispersed by the global circulation. The Chelyabinsk meteor event on 15 February 2013 started with an enhanced aerosol loading in the 30-35 km altitude range, as seen in extinction measurements of the OSIRIS/ODIN satellite instrument (Rieger et al., 2014). OMPS instrument onboard the Suomi NPP spacecraft has detected an extinction enhancement of a factor 10 at 40 km altitude and a few km width. It has also monitored the motion of the associated ring-shaped plume in the mid-latitude stratosphere, over at least 3 months, through its dispersion and its sedimentation from 40 to 30 km altitude (Gorkavyi et al., 2013).

Such local concentration enhancements in the middle stratosphere, at least 5 to 100 times higher than background levels, have been occasionally detected by the balloon-borne Stratospheric and Tropospheric Aerosols Counter (STAC) having operated from 2003 to 2011 (21 flights) at various latitudes (Renard et al., 2008; Renard et al., 2010). Such enhancements were mostly detected in the middle stratosphere. Although the instrument has not been designed to distinguish between liquid and optically-absorbing particles in terms of sizing, particles up to 5 μm were detected in some enhancements, unlikely to be pure sulfuric acid aerosols at such altitudes. In particular, 8 STAC flights were conducted above northern Sweden from 2 August to 7 September 2009, showing a strong variability from one day to another of the aerosol content and transient enhancements in the middle stratosphere (Renard et al., 2010). The enhancements could be very local, i.e. of a few km in term of horizontal extent (Renard et al. 2008).

## 4.3 Size distribution

The size distributions of NSPs in the stratosphere are poorly estimated. Indications have been mainly obtained by optical counters and by in-situ collectors with analyses by electron microscopy at ground.

Hunten et al. (1980), combining modeling calculations of the meteorite ablation around 80 km altitude and airplane collected particles (Brownlee, 1978), has proposed a bimodal repartition of the solid material in the middle stratosphere. The particles below 0.1 μm could come from the descending smoke particles, while the largest particles could originate from interplanetary dust and meteoritic debris. Nevertheless, the real content of NSPs could be more complex.





In the lower stratosphere, different size distributions have been detected. In the
Arctic, the optically-absorbing particles can dominate for size below 0.3 µm (Baumgardner et
al., 2004), The rocket engine plumes measurements, for the two cases-studies in the 17-19.5
km altitude range, have shown a three modal distribution centered below 0.01 µm, around
0.1 µm and around 2 µm (Ross et al., 1999). The fine volcanic ashes in the lower tropical
stratosphere after the Kelud eruption have been estimated to be below 0.6 µm in optical
diameter (Vernier et al., 2016). The PALMS instrument, which has detected two families of
NSPs in the lower stratosphere, has shown that the particles have an aerodynamical
diameter below 1 µm (Murphy et al. 2014). However, the instrument works only in the 0.2-2
µm size range and the aerodynamical diameter can significantly differ from the optical
diameter used by the optical aerosol counters.
The particles collected from airplanes present a wide variety of shapes and sizes. Soot
particles are in general smaller than 1 µm, in a compact or chain-like shapes (Pueschel et al.,
1992; Blake and Kato, 1995; Strawa et al., 1999). Submicronic to 10-µm particles of very
different shapes and natures were collected by Ebert et al. (2016) in the stratospheric polar
vortex at altitudes up to 21 km (Figure 8). Large particles, from a few µm to a few tens of
µm, are also present in the NASA aircraft JSC dust collections (Jessberger et al., 2001;
Sandford et al., 2016) and in the balloon-borne dust collection by the DUSTER instrument
(Ciucci et al., 2008). The concentrations of particles greater than a few µm could be in the
$10^{-6}$-$10^{-3}$ particles cm$^{-3}$ range.

**4.4 Particles chemical composition**


In the lower stratosphere, the presence of an aerosol population with mixed sulfuric
acid and metals, principally Fe, Na, K, Al, Cr, Ni atoms, has been derived from airborne mass-
spectrometry observations (Murphy et al., 1998; 2014). Soot or BC particles have been also
detected, in agreement with other observations (Blake and Kato, 1995; Strawa et al., 1999).
Also, Scott and Chittenden (1993) have shown from collected particles that the composition
of particles below 0.2 µm strongly differs from that of the common pure sulfate aerosols;
the main population of these ultrafine particles is composed of carbon, with traces of S, Na,
metal sulfates and chlorides.
The analysis of hundreds of airplane-collected NSPs in the polar winter lower
stratosphere by Ebert et al. (2016) has shown that refractory particles greater than 0.5 µm
mostly consist of silicate, silicate/carbon mixtures, Fe-rich, Ca-rich, and complex metal
mixtures including aluminum. The detection of metallic spheres within some mixed particles
might indicate the effect of high temperatures during the formation process but is not in line
with the results from airborne mass spectrometers which might rather suggest diluted Fe
atoms (Murphy et al., 2014). On the other hand, particles below 0.5 µm are mostly
composed of soot (Ebert et al., 2016). Most of these submicronic carbonaceous refractory
particles are completely amorphous and only a few particles are ordered with graphene
sheets (Schütze et al., 2017); minor traces of Si, Fe, Cr, Ni are often found in these
carbonaceous particles, and no difference has been highlighted in terms of size,
nanostructure and elemental composition for such particles either collected inside and
outside the polar vortex at least for the period of these observations, i.e. winter 2010.

*Figure 8: Examples of particles collected from an airplane (Figure 3 of Ebert et al. 2016),*
*showing their diversity in size, morphology and composition. The electron-microscope images*
*present refractory particles of (a) silicate spheres, (b) Fe-rich particle; (c) complex metallic*
*mixture (Al / Cr / Mn / Fe), (d) Ca-rich particle, (e) carbon /silicate mixture, and (f) silicate*
*particle*



Amongst the huge number of particles collected by the NASA airplane in the lower
stratosphere, particles assumed to be chondritic porous interplanetary dust, also named CP
IDPs, are composed of optically black aggregates of submicron components, with a wide
range of porosities (Rietmeijer, 1998). They are significantly enriched in carbon, most likely
in the form of pristine and complex organic molecules (Thomas et al., 1993; Flynn et al.,
2013; Koschny et al. 2019). Similarities between these particles and cometary dust particles
have been progressively suggested by analyses of samples from the Stardust flyby mission at
comet 81P/Wild 2 (Ishii et al., 2008), and more recently established through results on the
composition and the physical properties of cometary dust from the long-duration Rosetta
mission with comet 67P/Churyumov-Gerasimenko (Levasseur-Regourd et al., 2018; Mannel
et al. 2019). Also, the carbonaceous micrometeorites collected in Antarctica, also named
UCAMMs for Ultra Carbonaceous Antarctica Micro-Meteorites (Engrand and Maurette,
1998), after partial survival to the atmospheric entry and thus short transit in the
stratosphere, are estimated to be of cometary origin with up to 85% of organic matter in
volume (Nakamura et al., 2005; Dartois et al., 2018; Levasseur-Regourd et al., 2018) mixed
with tiny flakes of minerals.
In the middle stratosphere, Testa et al. (1990) have found Cl, S, Ti, Fe, Br, Ni, Zr, Zn,
Sr, and Cu elements to be present in the aerosols (elements having atomic number lower
than 16 could not be detected in their analysis), with 2/3 of the 23 analyzed non-graphitic
particles ranging from Al rich silicates to almost pure Fe, and one particle consisting almost
exclusively of Ba and S. The DUSTER collection has shown also two 100 μm spheres with O-
Si-Na-Mg-Ca composition (Ciucci et al., 2008) and the presence of pure carbon particles,
aggregates of CaCO and $CaCO_3$ grains (Della Corte et al., 2013).

## 5. Sources from ground to space

### 5.1 Context
All the information derived from the previously described reported studies appears
difficult to reconcile in terms of NSP concentrations, size distributions, and compositions. In
fact, most of the measurements could represent snapshots on specific geophysical
conditions. The strong vertical, temporal and typology variabilities of NSPs detected by LOAC
from one flight to another over a 6-year period seem to confirm the complexity of the
stratospheric NSP content and the difficulty to propose a global view of particles' origins.
Multiple sporadic and permanent sources of NSPs must be considered, coming from
Earth (emitted from the surface or produced within the atmosphere) and from space. In
particular, some authors have mentioned (disintegrated) meteoritic material, which is
indeed a source, but did not always consider porous carbonaceous interplanetary dust
particles as mainly originating from comets on prograde orbits. The grains resulting from
meteoritic disintegration could differ in size, shape and composition of those coming from
the interplanetary dust cloud. The recent results of the Rosetta mission on comet
67P/Churyumov-Gerasimenko have provided a ground-truth for such particles.
We present below the different sources of the NSPs from ground to space that can be
found within the stratosphere. The table 1 summarizes the main characterizes of such
particles.




| Nature | Typical size | Source | Origin |
|---|---|---|---|
| Volcanic ashes | < 0.6 μm | From ground | Natural |
| Biomass burning | < 1 μm | From ground | Natural |
| Pollution | < 1 μm | From ground | Anthropogenic |
| Polymeric nanocomposites | > 1 μm | Produced in the atmosphere | Natural |
| Rocket exhaust plume | < 5 μm | Produced in the atmosphere | Anthropogenic |
| Airplane soot | < 1 μm | Produced in the atmosphere | Anthropogenic |
| Meteoritic disintegration | All sizes | From space and produced in the atmosphere | Natural |
| Satellite disintegration | All sizes | From space and produced in the atmosphere | Anthropogenic |
| Interplanetary / cometary dust | < 1 m | From space | Natural |

*Table 1: Summary of the various sources of NSPs*


**5.2 Sources from Earth's surface**

The presence of volcanic ashes in the stratosphere associated with some volcanic
eruptions is due to an explosive process injecting directly the material into the stratosphere.
No mechanism of injection of ashes in the troposphere with subsequent transport to the
stratosphere by the Brewer-Dobson circulation has been reported so far.

Major biomass burning and organic fuel burning, having natural or anthropogenic
origin, are more frequent and can produce thick clouds of carbonaceous particles that can
reach the tropopause level. These particles can reach the lower stratosphere through direct
injection by cross-tropopause pyroconvection events or through transport of fire plumes
associated with overshooting convective systems (Damoah et al., 2006; Fromm et al., 2005;
de Laat et al., 2012). Also, they might be injected directly in the stratosphere through the
tropopause folds at tropical latitudes.

In addition to these sporadic events, periodic atmospheric mechanisms can consist in
a source of NSPs at a global scale. In particular, the Asian summer monsoon and the
associated Asian Monsoon Anticyclone (AMA) largely determine the composition of the
Upper Troposphere / Lower Stratosphere (UTLS). An accumulation of aerosols has been
pointed out inside the AMA and is present each year from June to September in the ~15-18
km altitude range in the UTLS region. This layer, known as the Asian Tropopause Aerosol
Layer (ATAL) (e.g. Vernier et al., 2018), is likely to be associated with Asian emissions of
anthropogenic pollutants like sulfur dioxide and volatile organic compounds, building a
population of NSPs consisting of a mixture of sulfates and organic material both as primary
and secondary organic aerosols. The ATAL is sustained by the convective activity of the Asian
monsoon as indicated by global model simulations (Yu et al., 2015; Fadnavis et al., 2017).
However, the precise composition, variability, trend and budget of the ATAL are still largely
uncertain and are currently under investigation. After the breakup of the AMA, the signature
of the ATAL is detectable on the extratropical aerosol budget in the northern hemisphere



(Khaykin et al., 2017) indicating that combined processes, i.e. emissions in Asia, convective
activity and general circulation, impact part of the global stratospheric NSP population.
**5.3 Production within the atmosphere**
If soot particles emitted from airplane engines can be directly injected in the lower
stratosphere during their cruise, their contribution is expected to be low with respect to
other sources coming from natural and anthropogenic biomass burning (Baumgardner et al.,
2004; Hendricks et al., 2004; Schwarz et al., 2006), although the air traffic is increasing.
Hypothetical long-lived volcanic soot particles could be also produced in the
stratosphere due to thermal decomposition of methane in the volcano eruption column
(Zuev et al., 2014, 2015), but this process need further studies and confirmation.
The rockets exhaust and the disintegration of satellites subsequently to their entry in
the Earth's atmosphere produce locally alumina, hydrocarbon and metallic debris (Ross et
al., 1999; Cziczo et al., 2002. The local content of these specific refractory NSPs increases
whenever measurements are fortuitously conducted inside a plume (Newman et al., 2001).
Such particles can be also collected far from their sources at different times of their transit in
the atmosphere during which their physical properties are likely to evolve. As a result, some
of them have been found to be mixed with particles having other origins (Ebert et el., 2016).
Compact particles and filaments having 5-10 μm in size and up to several mm long,
composed of carbon polymeric nanocomposites, can be produced in ambient air inside
plasmas (Hamdan et al., 2017). Such conditions occur during atmospheric entries of
meteorites and satellites/rocket debris and during storm lightning (Courty and Martinez,
2015) and perhaps during high-energy phenomena in the stratosphere such as blue jets and
sprites. These nanocomposites are characterized by the presence of Fe-Ni-Cr elements.
Some of the grains collected by Ebert et al. (2016) present the same composition and could
be nanocomposites produced within the atmosphere instead of meteoritic material. Also,
dusty plasma spherical particles typically in the 0.1-1.5 μm size range can be produced in
glow discharge (Pereira et al., 2005) as those encountered in the atmosphere. Layers of such
particles (spheres and filaments) could produce significant local concentration
enhancements of micron(s)-sized aerosols.
The rare events of the disintegration of large meteoroids, with sizes above about 10
m, occur several times per century and can produce a large plume of dust that can take
months to sediment, as for the Chelyabinsk meteor in February 2013. Disintegration of
meter-sized meteoroids is detected several times per year and can produce local layers of
dust. And partial or total disintegration of cm-size meteoroids could occur daily. The
disintegration altitude depends on the velocity entry, the incidence angle of the trajectory,
the density and the composition of the meteoroid (minerals like olivine, iron, ices, complex
organics; e. g. Fortov et al., 2013; Coulson et al., 2014). The probability of crossing such a
layer fortuitously during in-situ measurements is very low but non-negligible.
Meteoritic ablation may begin by altitudes of about 180 km.  Layers of minerals and
metals are then present in thermosphere between 100 and 80 km altitude from the
recondensation process (Rapp et al., 2007; Bardeen et al., 2008; Plane, 2012) that produce
nanometer-sized smoke particles (Antonsen et al., 2017). Such NSPs need to be aggregated
or to grow through the condensation of sulfuric acid when transported downward to
produce particles of at least of 150 nm to be optically detectable by optical aerosol counters.


**5.4 Dust from space**
The Earth orbits around the Sun within the interplanetary dust cloud, which is a wide
and flattened circumsolar cloud built of dust particles (e.g. Koschny et al. 2019). Their sizes
range from a few tens of nm to a few decimeters, with dominant sizes around hundreds of
µm. The spatial density of the interplanetary dust cloud increases towards the Sun and its
near-ecliptic symmetry surface, although it remains extremely low, with about 5 to 20
particles of about 10 about µm size per $km^3$ in the vicinity of the Earth (e. g. Levasseur-
Regourd et al., 2001).
Since particles within such a size range slowly spiral towards the Sun (under Poynting-
Robertson effect), the existence of the interplanetary dust cloud indicates that a more or
less continuous replenishment takes place. Interpretation of observations in the visible and
infrared domains (Lasue et al, 2007; Rowan-Robinson and May, 2013) and dynamical studies
(Nesvorny et al., 2010) indicate that most of the interplanetary dust particles reaching the
Earth's vicinity are of cometary origin, with a contribution of about 85% of the total mass
influx from short-period comets with a prograde motion, called JFCs (Jupiter Family Comets).
Dust particles of cometary origin have long been understood to form meteoroid
streams, such as the August Perseids from comet 109P/Swift-Tuttle. Besides, infrared
observations have allowed the discovery of faint dust bands, attributed to collisions within
the asteroid belt (Dermott et al., 1984) and of narrow and elongated structures (so-called
dust trails) along JFCs (Sykes et al. 1986; Reach et al. 2007), produced by the ejection of large
dust particles or peebles from cometary nuclei.
Because of its sources of replenishment, the interplanetary dust cloud is not a
featureless structure (e. g. Levasseur and Blamont, 1973), although dust particles freshly
injected are progressively randomly distributed into the cloud. Whenever interplanetary
dust particles impact the atmosphere, they induce the formation of sporadic meteors,
meteor showers, or even meteor storms (from fresher dust particles in the cloud) that
present a strong temporal and spatial heterogeneity. The flux of entry particles can be tens
to hundreds of times higher than during background conditions during the main permanent
meteor shower episodes; four main events may be mentioned, the Quadrantids (beginning
of January), the Aquarids (beginning of May), the Perseids (mid-August) and the Geminids
(mid-December)
As already suspected from comet 1P/Halley flybys in 1986, and established by the
Rosetta mission with 67P/Churyumov-Gerasimenko in 2014-2016, the refractory component
in comets is rich in high molecular-mass organics, possibly about 45% in mass and 70% in
volume (Bardyn et al., 2017; Herique et al., 2017; Levasseur-Regourd et al. 2018); the flux of
exogenous material from interplanetary dust entering in the Earth atmosphere is thus rich in
complex organics. The dust particles ejected from the nucleus of 67P/Churyumov-
Gerasimenko are porous aggregates covering a wide range of sizes, at least from tens to
hundreds of µm (Merouane et al., 2016), and of porosities with likely fractal structures
(Langevin et al. 2016; Mannel et al. 2016; 2019). Considering the prevalence influx of dust
particles originating from JFCs, the speeds of interplanetary dust particles impacting the
Earth's atmosphere are mostly low, around 15 km/s range (Nesvorný et al., 2010) or less.
Both the relative velocity and the morphology of these porous dust particles should enable





the survival of significant amounts of cometary organics within the atmosphere (Levasseur-
Regourd and Lasue, 2011).
The total amount of material entering the Earth atmosphere is still not well
determined, with a daily value in the 5-270 tons range depending of the various techniques
used for the determination (Plane, 2012); a mean value of about 100 tons per day is
frequently assumed (e.g. Rietmeijer, 1998). Carrillo–Sánchez et al. (2016) have considered
the cosmic spherule accretion rate at South Pole, the Na and Fe flux measured in the upper
mesosphere coming from the ablation of the incoming solid material, and the satellite
measurements of the interplanetary dust cloud radiation. Using ablation-modeling
calculations, they have proposed that dust coming from the JFCs contribute to about
80±17% of the dust mass entering the Earth atmosphere estimated at 43±14 tons per day. A
large fraction of NSPs commonly *assumed to have a meteoritic origin could in fact originate*
*from comets. Thus, it can be suggested* that the aggregated porous and complex organics-
rich particles detected in the stratosphere (Ebert et al., 2016; Schutze et al., 2017) could
come from comets, not from asteroids, while silicates-rich particles could come either from
comets or asteroids.
Nevertheless, two difficulties may arise for the detection and the identification of the
incoming material and the identification of their origin (cometary dust or meteoroid
disintegration in the atmosphere). First, some dust particles may be broken during their
atmospheric travel, producing a significant number of smaller particles, some of them being
not detectable. Secondly, carbonaceous particles might originate either from space or from
Earth. Thus, some layers of concentration enhancements could be produced for particles
with similar physically properties as the terrestrial ones but originating from space.

## 6. Origin of stratospheric concentration enhancements and transport mechanisms

We discuss here how LOAC particle counter observations under weather balloons
could help to better understand the origin and the transport of the NSPs leading to
concentration enhancements in the stratosphere.
The enhancements reported by various authors were observed sometimes during
well-known meteor shower periods but also apart from these events. As said before, about
25% of the 135 LOAC and 21 STAC flights exhibit strong enhancements. Such a probability
seems to be too high for fortuitous detections of meteoritic disintegrations or of
fragmentation of large interplanetary dust.
Figure 9 presents the number of concentrations enhancements per flight per week
for all the STAC and LOAC balloon flights as a function of the day of the year. The main
meteor showers are represented by dotted lines (the thickness of lines is related to the
intensity of the episode). No obvious correlation is statistically detected between the
variability of the number events and the meteor shower dates and intensities, although
some fortuitous coincidence can exist. Then, it can be concluded that the concentrations
enhancements in the stratosphere are not directly related to the meteor showers.

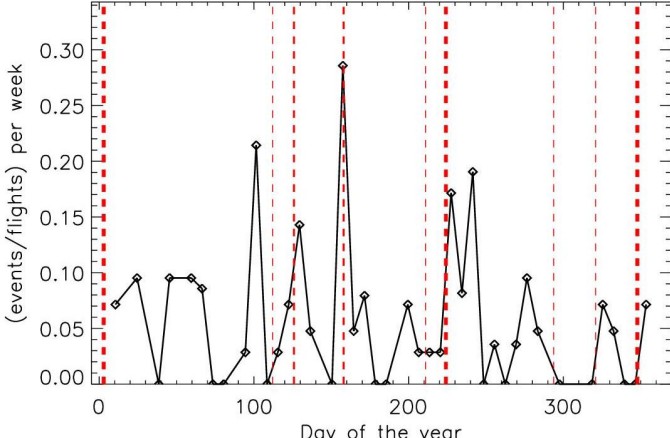

*Figure 9: Number of concentrations enhancement per flight per week, for all the LOAC and*
*STAC measurements during balloon flights; the red dotted lines represent the main meteor*
*showers episodes (the thickness of the dotted lines is related to the intensity of the episode)*

Different dynamical processes may explain the concentration enhancements. A
process differing from the common atmospheric transport mechanisms, the gravito-
photophoretic effect (Rohatschek, 1996; Pueschel et al., 2000) could allow light-absorbing
particles (like BC) to be lifted until the gravity counteracts their ascent, which could explain
transport of carbonaceous particles higher up in stratosphere, as for the 2017 Canadian
wildfires (Ansmann et al., 2018; Haarig et al., 2018). Such carbonaceous aerosols could stay
for several months at the hemispheric scale for these specific events (Kloss et al., 2019) or be
present ubiquitously in the tropical lower stratosphere (Murphy et al., 2007; 2014).
For particles of any nature, as proposed by Beresnev et al. (2012), vertical winds in the
stratosphere could be the basic force mechanism rather than gravito-photophoresis for the
formation and the spatial and temporal stability of aerosol layers (Gryazin et al., 2011). The
vertical winds could provide a levitation of the particles in the stratosphere and could form
spatial dynamical traps, in which aerosols are compelled to be in form of thin layers. The
averaged vertical wind could be the strong competitor of turbulent diffusion that prevents
from the stratification of stratospheric aerosols.
Another process could provide the levitation and the accumulation layers of soot and
BC particles: the radiometric photophoresis effect. Improvements in the theoretical works
show that accumulation layers of soot and BC particles could exist at different altitudes in
the stratosphere, depending on the size and the shape (compact or fractal) of the particles
(Beresnev et al., 2017). The larger the particles are, the potentially higher the accumulation
layers could be. In a steady-state atmosphere, accumulation layers of submicronic soot can
be in the 10-30 km altitude range; soot aggregates of several micrometers could reach 60 km
altitude. But in a non-steady atmosphere, we may expect that these layers are dynamically
perturbated and then can disappear.
Finally, strong gravity-wave events could also locally increase the aerosols content
and can produce thin layers, as observed one time at mid-latitude with the balloon-borne
LOAC (Chane-Ming et al., 2016).





Some of these phenomena could explain most of the concentration enhancements
attributed to NSPs and previously detected by the STAC balloon-borne aerosols counter at
different altitudes and locations (Renard et al., 2008, 2010), and also for the sparse
enhancements of carbonaceous particles observed by CALIOP/Calipso during the monsoonal
convective season over India (Govardhan et al., 2017). Also, they could explain the enhanced
concentrations layers detected after Leonids meteor shower events, linked or not to the
disintegration of incoming extraterrestrial material. Soot particles can have different size
distributions and fractal/compact shape depending on their origin and their aging (e.g.
Adachi et al., 2007), which could explain the various altitudes for the detected accumulation
layers.
A statistical analysis of the altitude of the concentration enhancement events
detected by LOAC seems to indicate a double repartition, one centered at around 17 km and
the second one at around 30 km (Figure 10). The analysis is conducted by calculating the
percentage of events in respect with the total number of measurements available at the
various altitudes. This double repartition looks like the one proposed by Beresnev et al.
(2017) for the accumulation layers of fractal and spherical carbonaceous particles
respectively (which could correspond to porous fractal aggregates and dense aggregates).
The LOAC typologies indicate that optically-absorbing particles dominate the aerosol-
enhanced layers although sulfates are also present. The origin of these particles is unknown,
since carbonaceous particles coming from space, produced within the atmosphere and
emitted from the Earth's surface can be compact and/or fractal.

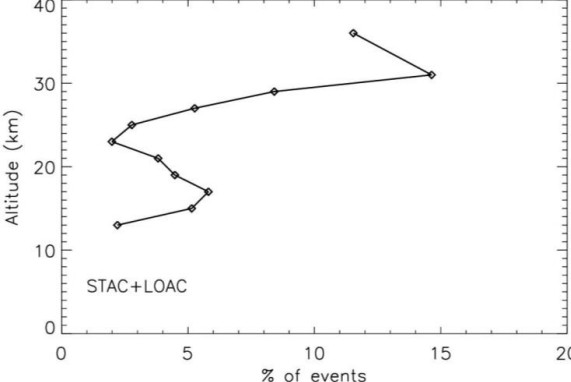

*Figure 10: Evolution with altitude of the percentage of concentration enhancement events*
*detected by the STAC and LOAC aerosol counters during balloon flights*

## 7. Background concentrations for large NSPs

Finally, LOAC can be used to estimate the background content of large NSPs, with
sizes ranging from several μm up to 100 μm, usually expected to come from space
(interplanetary dust and meteoritic debris), with possible growth by sulfuric acid
condensation consequently to their descent in the stratosphere (Bardeen et al., 2008).
Considering the number of flights per year, the balloon ascent speed (~5 m.s$^{-1}$) and
the pump flow (~2 L.min$^{-1}$), LOAC has sampled about 1 to 2 m$^3$ of air per year in the 15-35



km altitude range. Figure 11 presents the mean evolution of the concentrations with
altitudes of particles detected for the mid-2013 – mid-2019 period, for three size classes (5-
7.5 µm, 10-17.5 µm, 20-100 µm) in layers of 5 km width; the error bars represent the
interannual variability. The concentrations seem to decrease by about of factor two from 15
to 35 km for the three size classes. In the middle stratosphere, the mean concentrations of
particles having sizes of about 5 µm, 10 µm and 20 µm are of about 50, 10 and below 10
particles m$^{-3}$, respectively. Although the annual volume of air sampled by LOAC is low, it
seems that a tendency could be tentatively pointed out. The lower concentrations values are
in the 2017-2018 period (with no particles greater than 10 µm in the middle stratosphere),
while the higher concentrations values are in the 2015-2016 period.
Ebert et al. (2016) have collected during the RECONCILE campaign inside the polar
vortex about $10^3$ particles m$^{-3}$ greater than 3 µm in the lower stratosphere (below 21 km).
On the other hand, Hunten et al. (1980) using the Brownlee (1978) measurements have
estimated the concentration of interplanetary dust (or micrometeorites) at 30 km to be of
about $10^{-3}$ particles m$^{-3}$ for sizes greater than 5 µm. Finally, the concentration of collected
particles greater than 5 µm by the DUSTER instrument during one balloon-flight at 38 km
was of about one particle m$^{-3}$ (Della Corte et al., 2013). The LOAC estimates cover all these
values. Although these sparse measurements have sampled a small volume of the
stratospheric air, they could indicate a strong variability for the large particle's
concentrations.

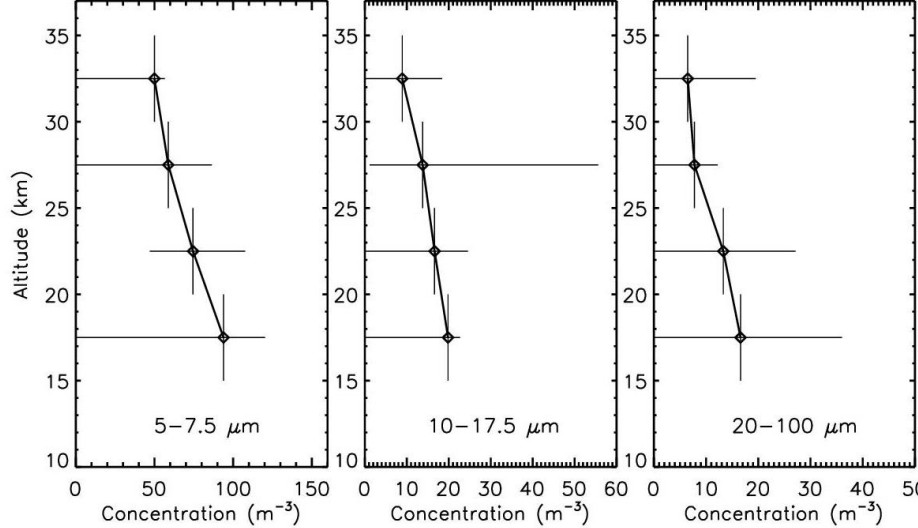

*Figure 11: Mean altitude evolutions of the concentrations of the large particles detected by LOAC for the 2013-2019 period, for 3 size classes; the errors bars represent the interannual variability*

Some of these particles are often classified as "meteoritic material", based on the
presence of Fe, Mg, Ni, Cr, Na in their compositions derived from collected samples or from
in situ analysis (Cuicci et al., 2008; Ebert et al., 2016). Interplanetary dust, mainly originated
from Jupiter Family Comets, must be also considered. Porous cometary dust particles with


not too high relative velocities happen to reach the stratosphere (Levasseur-Regourd et al.,
2018) and about 10% of the interplanetary dust flux is likely to reach the Earth's surface
without suffering any melting (Dobrica et al., 2010); thus, they can contribute to the
carbonaceous particles found in the stratosphere. Nevertheless, we may conclude that the
presence of interplanetary material in the stratosphere is dramatically overestimated in the
litterature, since the detected concentrations are several orders of magnitude above the
estimated concentration of material in space at Earth orbit, i. e. $10^{-14}$ particles cm$^{-3}$ for size
greater than 10 μm (e. g. Levasseur-Regourd et al., 2001). This last value might be tens or
hundreds higher during meteor shower events and even more in case of meteoritic
disintegrations inside the stratosphere, but still without reaching the stratospheric
concentrations of large NSPs.
We can however propose several explanations for these discrepancies:
- The flux and the size distribution of incoming material from space is strongly
underestimated. In particular, the size distribution estimated from impacts on exposed
surfaces in space (Mandeville et al., 1991; Love and Brownlee, 1993; Kalashnikova et al.,
2000) could be inaccurate and the retrieved diameters and thus the size distribution could
differ from those obtained by optical instruments.
- Large fluffy particles (hundreds of μm or greater) entering the Earth atmosphere
could undergo breaking processes, producing large number of micron-sized particles and less
nanometer-sized (smoke) particles than expected during the ablation processes.
- A space-time sampling bias can be assumed. In other words, the dates and the
locations of the in situ measurements could be not fully representative of background
conditions, in particular where the regular flights are not well evenly time spaced.
- Some particles produced by the Chelyabinsk meteor disintegration could remain
longer than expected in the stratosphere. This could (partly) explain some high
concentrations values detected by LOAC since the beginning of the measurements in 2013
until the return to lower values at the beginning of 2017; the low concentration of big
particles subsequently to the dilution of the Chelyabinsk meteor cloud in the stratosphere
could be undetectable by spaceborne remote-sensing observations.
- The instruments may have mainly detected material coming from the Earth during
specific events or produced inside the atmosphere and transported to the middle
stratosphere; these particles can remain in some accumulation layers (Beresnev et al., 2012,
2018) and even can agglomerate. We are in favor of this last explanation, since the
concentrations decrease with increasing altitudes, and the accumulation layers were indeed
detected by the STAC and LOAC aerosols counters at the altitude range predicted by the
modelling calculations. Also, the identification of the origin of the particles based on their
composition only could be inaccurate. As an example, the nanocomposites produced inside
the atmosphere (Courty and Martinez, 2015) could have chemical elements like those found
in meteorites or in interplanetary dust.

## 5. Conclusions

**5. Conclusions**
Even if a large variety of in situ and remote sensing measurements have been
conducted under different atmospheric conditions, none of them can provide a
comprehensive view describing the whole complexity of the stratospheric NSP content. The
sources are multiple and most of them are non-permanent. One can expect a strong


variability of the chemical composition, the size and the concentration of stratospheric
particles from one session of measurements to another. Also, the particles are likely to be
detected or collected at different stages of their life cycle from their emission to their
removal from the atmosphere during which they are transformed (e.g. condensational
growth, coagulation, inclusion). This complicates the determination of their origin, of their
physical properties and of the processes controlling the evolution of their size distributions
and concentrations.
The LOAC balloon-borne optical counter has contributed to better understand the
origin of the complexity of the NSPs content. It has confirmed the presence of enhanced
layers in terms of concentration of submicronic, and sometimes larger, particles in the lower
and middle stratosphere. The six years of regular flights (2013-2019) have shown a strong
temporal variability of such events, which does not seem to be correlated to the main
meteor shower events. At present, the more plausible hypothesis for such spatial and
temporal behavior is the presence of accumulation layers of terrestrial and atmospheric
particles due to the dynamical and photophoretic effects.
Frequent LOAC balloon flights have shown the necessity to often conduct new
systematic stratospheric measurements at various locations to answer the following open
questions:
- How to distinguish between the various sources of particles? How to determine
their average percentages in terms of size, number concentration and mass, and their
evolutions with altitude?
- Are there specific physical and chemical markers allowing to distinguish black
carbon and soot particles on a way of formation, types and origins?
- Do most of the complex organic particles are coming from space (e. g. cometary
particles) or from the Earth?
- Is there an evolution of the physical and chemical properties of the NSPs in the
stratosphere (i.e. aging)?
- Is there an increase of the BC/soot particles content in the stratosphere due to
anthropogenic activities especially in Asia?
- How long is the residence time of NSPs, depending on their origin and the different
levitation processes involved?
- Are the local concentration enhancements really accumulation layers coming from
the vertical winds and photophoretic processes?
- Is there a direct link between (charged) NSPs and high-energy phenomena above
thunderstorms?
- Finally, what are the consequences of the presence of such particles on Earth's
radiative balance and on the stratospheric chemistry?
Current aerosols counters cannot be used alone to answer these questions, thus new
light instrumentation will be needed, using for instance mass-spectrometers and collecting
devices for frequent and low-cost flights. Also, an optical instrument performing
measurements at several scattering angles could be useful to better evaluate the mean
composition of the particles, as done for example at ground with the laboratory instrument
PROGRA2 (e. g. Hadamcik et al., 2007; Francis et al., 2011). Long-duration balloons flights
from weeks to months with daily vertical excursions and carrying a poly-instrumented
gondola could be also useful to better evaluate the temporal and spatial variability of NSPs
in the stratosphere. Aerosols counter and mass spectrometer can be part of the gondola, but
also a possible future instrument that can derive the isotopic composition of the solid





material to better distinguish between the various sources, as proposed by Kalashnikova et al., 2016; Beresnev and Vasiljeva, 2018 for carbonaceous particles.

Remote-sensing measurements from future satellite platforms, like the EarthCARE spaceborne lidar (Illingworth et al., 2015), coupled with balloon-borne measurements, could help to better identify the various natures of stratospheric aerosols and their variability. Finally, a new in-situ counting instrument along the Earth's orbit could be proposed to better estimate the size and the concentration of incoming material from space.

Such improved knowledge of the stratospheric aerosols and the role of the NSPs will be useful for improving chemistry and climate modeling works, including radiative transfer calculations over the whole atmosphere.

**Author contribution:** Jean-Baptiste Renard designed the LOAC experiment and processed the data. Gwenaël Berthet and Damien Vignelles participated in the improvement of the instrument and of the data processing, and in the data interpretation. Anny-Chantal Levasseur-Regourd, Sergey Beresnev, Alain Miffre, Patrick Rairoux and Fabrice Jégou participated in the analysis of the origin of the aerosols and of their spatial and temporal variability.

**Acknowledgments**. The LOAC instruments were funded by the French Labex "Étude des géofluides et des VOLatils–Terre, Atmosphère et Interfaces – Ressources et Environnement" (VOLTAIRE) (ANR-10-LABX-100-01) managed by the University of Orleans. The STAC and LOAC flights were funded by the French Space Agency CNES. We want to thank the CNES balloons launching team at Aire sur l'Adour, the MeteoModem Company for the flight for Ury (France), and Nelson Bègue and the LACy for the flights at Ile de la Réunion. We want to thank Marie-Agnès Courty for information concerning the local production of nanocomposite in the atmosphere, Andrei Vedernikov for fruitful discussion, and finally CNES for its support in the scientific analysis *of* Rosetta data. Sergey Beresnev want to thank the Ministry of Science and Higher Education of the Russian Federation, the research project #3.6064.2017/8.9.

The STAC data are available at:
https://cds-espri.ipsl.upmc.fr/etherTypo/index.php?id=667&L=1
The 2013 LOAC data are available at:
http://mistrals.sedoo.fr/?editDatsId=1017&datsId=1017&project_name=ChArMEx
The LOAC data from 2014 are available at:
https://cds-espri.ipsl.upmc.fr/etherTypo/index.php?id=1699&L=1

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
