# Peer review of "1The complex origin and spatial distribution of non-pure sulfate2particles (NSPs) in the stratosphere"

_Atmospheric Chemistry and Physics, 2019_

## Referee Comment (RC1) · Anonymous Referee #1 · 19 Dec 2019

The complex origin and spatial distribution of non-pure sulfate particles (NSPs) in the stratosphere, by Jean-Baptiste Renard et al.

Referee report

General Comments

Starting as a nice overview of the literature on stratospheric absorbing aerosol, this paper becomes much less convincing from the moment that experimental results are analysed, and mainly interpreted. The authors often provide personal interpretations as if they were evidences, without the necessary care. Their references to the literature is also sometimes suffering from a biased or incomplete report on what is written in the

cited papers. The authors lump together carbonaceous aerosols, black carbon, meteoritic smoke particles, and particle containing non-volatile residues. Further (Section 5.3), they put the emphasis on unusual processes without any reference to their real impact or importance. They give a perception of "huge amounts of absorbing particles" without real distinction amongst them, while there are large differences in optical and microphysical characteristics e.g. between a sulfate aerosol droplet that grew on a non-volatile condensation nucleus (possibly of meteoretic), soot particles from anthropogenic origin, or meteoritic smoke particles. The general view they are giving on stratospheric aerosols is sometimes very chaotic, and in some cases, biased and not in accordance with the literature they are citing. Nowhere, the authors give any perspective about the exact importance of all kinds of absorbing particles with respect to the entire stratospheric aerosol population.

Furthermore, the style of this paper alternates between a review paper (sections 2, 4, 5) and the presentation of personal work around LOAC (sections 3, 6, 7), for which the authors do actually not provide any serious analysis nor any solid, rigourous, and convincing results. LOAC results presented here have a particularly poor information content. The article includes a lot of speculative interpretations without any attempt to validate them using some quantitative data, what is particularly pernicious: the risk is high that conjectures without any solid ground are further cited without too much care (as the authors sometimes do here) and become supposed pieces of truth after a few citations.

The authors should much more valorise this review effort by starting from the categories identifies in Table 1, and to analyse the different aspects (particle size, chemical composition, etc.) for each identified aerosol type. They should clearly separate this review and the LOAC results, and if they choose to use these results, they should propose a solid, grounded, in-depth and quantitative analysis. Anyway, this paper needs a lot more rigour, e.g. through appropriate quantifications of their statements.

Specific comments

L. 61: I suggest slightly modifying the formulation to take into account the influence of the volcanic eruptions mentioned in the preceding sentences, since such eruption (e.g. Nabro in 2011) can really bring temporarily the atmosphere out of this background situation. For instance, the authors could add "out of such volcanic events".

L. 73: Do the authors mean "pure solid or liquid particles"?

L. 105-108: This sentence is very similar to the sentence on L. 26-29, and in some extend, to the one on L. 1016-1018. The authors should avoid such redundancy. They can explain, for instance, that this section proposes an overview of studies on NSPs detection using all kind of platforms listed in the previous sentence.

L. 133: "(. . .) which backscatter light is well-known": what do the authors mean?

L. 151-152: This sentence might requires citations.

L. 160-161: This sentence requires also a citation. I guess the sensitivity to the complex refractive index depends on the measurement principle used in in the particle counter. Maybe the text should be qualified in this sense.

L. 162-164: This is actually a very general issue in aerosol retrieval from optical measurements, also by remote sensing.

L. 167-169: "The fraction (. . .) not composed entirely of (. . .) "is a strange formulation. I guess there are other possibilities than water and sulphuric acid to form volatile materials. The authors should thus qualify their sentence (e.g. "i.e. mainly water and sulphuric acid"). "The difference between both channels": difference of what? Please be more specific.

L. 172: "is used": I guess the list of measured parameters depend on the study and on the instrument. "Can be" might be more exact.

Section 2.5: Although it is suggested in L. 225-226 through the mention of the assumption on the refractive index, it would be worth to explicitly mention that satellite measurements in the UV-visible spectral range cannot provide information on the aerosol composition (contrarily to satellite measurements in the IR such as MIPAS).

L. 263-266: What do the authors mean? The use of a priori hypotheses on shape, composition and size distribution is necessary. Do the authors mean: "the impact of the use of a priori hypotheses"? Concerning the citation of Bourassa et al., 2012, the authors might usefully prefer Rieger et al., Stratospheric aerosol particle size information in Odin-OSIRIS limb scatter spectra, Atmos. Meas. Techn., 7, 507-522, 2014 where a discussion of the retrieval sensitivity to such parameters is discussed.

L. 268: For the completeness, I suggest that the authors also cite Kovilakam and Deshler, On the accuracy of stratospheric aerosol extinction derived from in situ size distribution measurements and surface area density derived from remote SAGE II and HALOE extinctions measurements, J. Geophys. Res. Atmos., 120, doi:10.002/2015JD023303, 2015. This paper presents an important correction brought to the processing of the optical particle counting measurements by the team of the University of Wyoming.

L. 322-323: The effect of porosity effects is less taken into account in size retrieval studies, and a citation should be required.

L. 328-332: The number and range of provided size classes (19 sizes classes in the 0.2-50 $\mu$m) is very large, and it might be useful to give some more insight on the uncertainty as a function of the size range. In addition, I recommend providing a reference where the uncertainty aspects of LOAC measurements are discussed.

L. 344-354: This requires a citation.

L. 369-371: Do the authors mean that a different kind of balloon is used at Aire sur l'Adour, or is it some kind of nomenclature used at this place or by CNES? Please clarify.

L. 375: The profiles in Figure 2 do not show monotonic decreasing profiles with altitude. Please be specific.

Figure 2: Are the points around a concentration range of 1.E-7-1.E-5 cm-3 significant, or are they under the detection limit of the instrument? In the first case, please justify with references to a paper describing the specifications of the instrument. In the latter case, I suggest to remove them from the plot and to adjust the scale for the sake of clarity.

Caption of Figure 3: In my opinion, the authors should give a more detailed description of the figure, and more particularly of the meaning of the coloured regions.

L. 380-383: This explanation seems related to Figure 2. It should thus come before the mention of Figure 3. Interpretation in L. 381-383 is highly speculative and not sufficiently scientifically grounded: I guess many other reasons might explain the presence of one large particle at that place and time. The authors should remove this statement lacking of evidence.

L. 399-403: This criterion is not clear. How do the authors measure the "background" used as reference to assess the "strength of the concentration". Are they comparing values of the size distribution at a same altitude? This needs to be specified. Do the authors mean that they try to find bimodal size distributions, or discontinuities within a size distribution?

L. 409-411: Again, the interpretations of the origin of these particles are highly uncertain and the authors should just be content with the mention that, in view of the speciation index, these particles are likely to be absorbing, in agreement with Figure 5. Examples of possible composition inferred from the database mentioned in L. 345 (e.g. carbonaceous particles) might be given, but with appropriate references to other works to support their conclusions. As shown here, only the character "transparent", "semi-absorbing" or absorbing" may be inferred.

L. 410-411: It is hard to distinguish something special between 8 and 12 km: For some particle classes ($\sim$< 0.5 $\mu$m) the vertical profile is increasing toward lower altitudes, for other ($\sim$0.5-3 $\mu$m), it present a peak at 8 km, and for larger particles (∼3-50 $\mu$m), it decreases toward lower altitudes with typical concentrations of ∼1.E-3-1.E-4 cm-3. Furthermore, these results are very different from results of cirrus size distribution at mid-latitude presented by Zhao et al, J. Am. Met. Soc. https://doi.org.10.1175/2010JAS3354.1, finding typical size distributions with a maximum concentration around 5 to 30 $\mu$m from airborne in situ measurements and remote ground-based measurements, while the size distributions found here between 8 and 12 km present a minimum around 10-40 $\mu$m. Hence, the authors should present convincing arguments to interpret these particles as cirrus clouds, or remove this sentence.

Figure 7: It is strange that, contrarily to Figure 5 where two clearly different groups of points, spread in different "speciation zones", suggest at first sight the present of two different modes, in Figure 7, all indicated points seem to belong to a same group, spread over the 3 provided zones and even out of them. This seems to suggest, either that LOAC measurements cannot analyse adequately such case, or that another particle type might not be represented on the plot. Also strange is the fact that the yellow zone is obviously different in Figure 5 and 7, while the red and blue zones look similar. What is the reason for this discrepancy?

L. 437-447: "As a result" of what? I really don't know from which element the authors can conclude that LOAC and STAC measurements detected similar concentration enhancements: so far, no single STAC measurement was presented! Concerning the "strong gradients" at the considered altitudes in Figures 4 and 6, there is indeed a sharp structure showing a clear change in the air masses. However, it seems that all vertical profiles roughly undergo a jump of a similar amplitude, possibly suggesting that mainly the total particle number density is changing, but that the shape of the particle size distribution is hardly affected. This would undermine the conclusion drawn in the present discussion. Therefore, it is very important that the authors provide such an illustration of the particle size density as a function of the particle size, for the considered altitude and for the closest altitude levels (say, 3 or 5 altitude levels around 18 km for Figure 4, and around 28 km for Figure 6). In this way, they could verify that

the size distribution is clearly different at the level presenting a strong gradient. Also, if the authors want to make some statistics about "strong enhancements" (L. 439-440), they should provide some quantitative criteria to define such observation. They should also provide uncertainty estimates to correctly assess the strength of the phenomenon. Overall, the conclusion that most of the concentration enhancements corresponds to a dominating population of NSPs is clearly premature.

L. 459-462: If many research works (including the ones cites by the authors) show the importance of meteoritic smoke particles, this statement is much too simplistic in a paper on "the complex origin" of NSP. Also, although the mentioned authors indeed study the role and importance of meteoritic smoke particles in the atmosphere (often in a much broader scope than this particular region around 40 km altitude), I did not find in these papers any statement that extinction enhancements at these altitudes are definitely due to meteoritic smoke particles. The authors should thus be much more cautious in their mentions of the cited papers, and they should qualify their statement (e.g. "extinction enhancements are likely associated", or "the dominating role of meteoritic smoke particles has been shown at these altitude" . . .).

L. 463-465: Detection of absorbing aerosol requires techniques able to sound the complex character of the index of refraction, through polarization effects or scattering properties. This is the case of instruments such as MISR, OMI or PARASOL, but GOMOS does not offer such kind of possibilities. Hence, if it is possible from GOMOS to detect extinction with a spectral behaviour unlikely to be due to sulphate aerosols, it is not possible to attribute unambiguously such signature to the presence of absorbing aerosols particles. Please qualify this statement.

L. 482-484: Although Baumgarder et al. (2004) indeed mentions this result, this statement is biased in absence of the complete information (e.g. Figure 4 in Baumgarder et al (2004)) because it gives wrongly the impression that absorbing particles are fully dominating the particle size distribution. This is not true and for particles larger than 0.3 $\mu$m, the non-light absorbing particle are largely dominating everywhere. The authors should provide a complete, non-biased information and specify how the whole size distribution behaves.

L. 485-487: "Similarly": What is similar? Please reword.

Section 4.2: In this section, the authors are putting all together all kinds of events from fires to meteoritic smoke and even a large-size meteor, all of them seemly assembled under a common characteristic of "spectacular event". I am not sure there is a real scientific interest to consider such a category "sporadic strong enhancements" without much common characteristics of origin, composition, geolocation, or any other common feature that might interest the reader.

Section 4.3: This section is equally a catch-all of a bit of everything. It becomes rapidly clear how the authors lump all kind of absorbing aerosols together. Starting from the case of "meteorite ablation around 80 km and airplane collected particles", the discussion of this case continues with "smoke particles", "interplanetary dust" and "meteoritic debris" and becomes "NSPs" in the next sentence. There is thus an obvious confusion between all these cases, although there is no reason at all that aerosol modes as different as smoke particles, soot particles or other NSPs (maybe sulphate aerosols that grew on some meteoritic condensation nuclei) have similar general size characteristics, since their origin, chemical composition and morphology are completely different. Overall, this section looks like a messy catalogue of numbers of which I am not sure about the benefit.

L. 641: "The detection of metallic spheres": Please cite the corresponding work.

Section 4.4: Is it right that the two first paragraph (L. 630-650) refer to carbonaceous aerosols of anthropogenic origin, while the third one (L. 660-675) concerns meteoritic particles? And what about the last paragraph (L. 676-682)? The effort of reviewing all these works and providing an insight into the wide range of compositions of these aerosol types is very valuable, but it would be useful to split this overview into aerosol types (e.g. soot from anthropogenic activities, meteoritic particles from extra-terrestrial

origin, and possibly particles from unknown origin).

L. 697-702: The interest of this isolated mention of "prograde orbits" is unclear. Further, it is difficult to understand why, just after mentioning the difficulty to categorize aerosol types, the authors seem to look for further sub-categories.

L. 714-715: This statement requires revision: although this question was widely debated in the community, it is likely the volcanic plume of the Nabro (13°N, 41°E, June 2011) reached the upper troposphere due to the eruption and was transported afterward to the stratosphere along pathways provided by the Asian summer monsoon.

L. 716: What do the authors mean by "organic fuel"?

L. 722; Tropopause folds are phenomena occurring in the extra-tropics.

L. 720: Which kind of overshooting convective systems do the authors have in mind? The three provided reference correspond to pyroconvection.

L. 725: "largely determine the composition of the UTLS": is it true that the AMA is the determining factor of the composition of UTLS? It might be necessary to quality this statement in "plays a major role in the composition (...)". This statement would require a reference in any case.

L. 734-735: This sentence requires some reference. In particular, the authors should mention important projects specifically dedicated to the study of the ATAL and its composition such as STRATOCLIM (Brunamonti et al, Balloon-borne measurements of temperature, water vapor, ozone and aerosol backscatter on the southern slopes of the Himalayas during StratoClim 2016-2017, Atm. Chem. Phys., 18, 15937-15957, 2018) or BATAL (Vernier et al., BATAL, the balloon measurements campaign of the Asian tropopause aerosol layer, B. Am. Meteor. Soc., May 2018, 955-973, http://doi.org/10.1175/BAMS-D-17-0014.1, 2018).

L. 750-756: How significant are such source in the stratosphere?

L. 757-768: What is the frequency of re-entry of satellites or rocket disintegration? The probability of observing such event looks particularly low to me, and the interpretation of Ebert's results in the light of Hamdan's work looks highly improbable and speculative to me

Section 5.3: It is a little bit strange that the overview of the NSP production mechanisms describes as second mechanism the formation of "hypothetical long-lived volcanic soot particles" and as third one "rockets exhaust and the disintegration of satellites" of which the relative importance is particularly uncertain in the global NSP production. This one is followed by "compact particles and filaments" possibly (hypothetically?) occurring during atmospheric entries of meteorites and satellites/rocket debris" , and "rare events of the disintegration of large meteoroids, with sizes above 10 m". Finally, the very last one mentions meteoritic ablation likely to be about the most important of all the cited processes. The authors should present the different mechanisms in the order of importance and impact on the atmospheric processes, starting from the most important. They also should quantify in some way their importance, in terms of concentration, spatial coverage, lifetime, occurrence rate, etc.

L. 795-797: Sentence is not clear.

L. 797: Interpretation may be arbitrary. Do the author mean "analysis"?

L. 804: "faint dust bands attributed to collisions within the asteroid belt": It is not clear if the authors refer to some particle "cloud", or to some spectral band. Please clarify.

L. 808-810: Odd sentence. Please reword.

L. 812-813: Which cloud? This sentence requires some reference. Idem for L. 813-817.

L. 826-828: Incorrect sentence: What is "prevalence influx"? Speed is a physical parameter, thus singular.

L. 829: "relative velocity": with respect to what?

L. 829-830: The authors misuse results published by Levasseur-Regourd and Lasue (2011). These authors draw their conclusions about the probable "survival" of significant amount of organics for seeding life on early Earth, during the Late Heavy Bombardment epoch, "while the spatial density of dust in the interplanetary dust clouds was orders of magnitude greater than nowadays".

L. 841-842: The meaning of the use of italic characters is unclear.

Section 5.3: The authors give a lot of details about interplanetary dust particles and discuss extensively the origin of these particles (comets or asteroids). However, the conclusions of interest for the study of NSP is particularly poor and vague: enhancements of interplanetary aerosol particles [of carbonaceous aerosols] might show similar properties (which ones ?) as aerosol particles of terrestrial origin. The authors should shorten this section.

L. 866-867, Figure 9: Does it mean that STAC and LOAC flights span several weeks? This is not what previous text suggests (e.g.:, Figures 1 to 7, corresponding to flight on a single day). The authors should clarify what they mean on Figure 9, and how many flights are concerned. A table with an overview of all types of flights and their characteristics might be useful.

L. 863-865: On which criteria do the authors base their statement? How fortuitous such event is depends on its persistence and frequency. Based on the potential total daily amount (up to 270 tons) given in L. 833, meteoritic smoke particles might be not so fortuitous.

L. 869-872: Did the authors calculate any correlation?

L. 871-872: I do not see any convincing, quantitative analysis that might lead to this conclusion.

L. 880-917: The authors list a full page of speculative explanations, without proposing any serious analysis on how their measurements may support any of them. I have

some doubt about the relevancy of such approach.

L. 918: "double repartition" is incorrect.

L. 921: "This double repartition looks like the one proposed by Beresnev et al. (2017)": this is again a very weak "analysis" referring to a non peer-reviewed proceeding, with a vague description ("the [double repartition] for the accumulation layers of fractal and spherical carbonaceous particles respectively"). Such a superficial analysis is totally insufficient.

L. 925-927: What does it mean? What do the authors want to prove?

L. 945-946: Following Figure 11 and taking into account the error bars, there is no statistically significant decrease of the concentration with increasing altitude.

Caption Figure 11: "Evolution" (should be singular) seems more a time-related concept. "Altitude dependence" seems more adequate. What is "mean evolution"? Please specify what the circles represent.

L. 948-949: Even if the ensemble of the considered measurements are likely to provide much useful information, it is impossible to conclude on any trend from the results given in Figure 11 (profiles and their error bars) and the fact that the total analysed volume of air represents about 1 to 2 cm3 of the whole stratosphere.

L. 949-951: What do the authors intend to prove by this statement? What is the available measurement sample and the representativeness of these observations?

L. 952-961: This "analysis" is inadequate. How do LOAC flights match with the REC-ONCILE campaigns inside the polar vortex? Even by summing the contributions to all panels of Figure 11, the result does not reflect in any way a concentration of about 1000 particles/m3 for particles greater than 3 microns below 21 km. And saying that LOAC measurements agrees with cited results at 30 km and 38 km respectively, is meaningless. The authors should remove all this discussion.

L. 973-975: This reference is difficult to find, and the mentioned rate of 10% is not mentioned in a summary found on internet. However, the title of the paper specifies that this work concerns micrometeorites collected at ground level in ultra-clean snow in Antarctica. This is far from representative of the interplanetary dust flux reaching the (global) stratosphere. The rest of the section (L.975-1011) is an additional hope of speculation without any attempt to test them against any quantitative result.

L. 1030-1034: The developments as presented in the paper do not allow drawing these conclusions.

L. 1072-1076: These sentences include generalities without any relevant added information. I suggest removing them.

Technical comments

L. 27: missing comma.

L. 35-36: "ranging between 17 and 30 km altitude"?

L. 227: "They cannot access the local variability" or "They give no access to the local variability".

L. 114: "sparse of time-series measurements": looks odd. Maybe "sparse or continuous", "sparse or routine measurements"?

L. 118: "abundance". Physical parameter, thus singular.

L. 127: "EARLINET".

L. 241-243: Odd sentence (combination of "often" and "in general").

L. 244-246: "such as volcanic eruptions, injection (. . .), or pyroconvection"?

L. 264: "hypotheses".

L. 313: "can be"?

L. 333: I suggest "The particle size provided by LOAC" to avoid the strange association "particles provided by LOAC".

L. 355: "corrected for".

L. 373-374: "an example of vertical (. . .)"; "on 17 August 2017"

L. 444, Figure 3 (and maybe elsewhere): "absorbing" instead of "absorbent".

L. 704: "main characteristics".

L. 752: missing ")".

L. 793: Please correct the sentence.

L. 795: "with' instead of "within".

L. 817: missing '.'

L. 971: "originating".

L. 1175: "boundary".

L. 1263: "Fadnavis, S.".

L. 1330: "Process."

L. 1595: "OSIRIS".

L. 1014: "Conclusions" is Section 8.

---

## Referee Comment (RC2) · Anonymous Referee #2 · 13 Mar 2020

The complex origin and spatial distribution of non-pure sulfate particles (NSPs) in the stratosphere, by Jean-Baptiste Renard et al. gives an overview of the literature on stratospheric aerosol. With more than 100 cited papers a lot of diverse information about stratospheric particles is compiled. In this "review"-part (chapter 1,2,4,5) specific results from diverse papers are mixed together to present an overall picture. But in some cases this presented overall picture is inaccurate as not all presented results can be mixed in reality because of a complete different data base in the cited papers (e.g. particle composition of small carbonaceous particles presented by Ebert et al., 2016 on the one hand and Schütze et al., 2017 on the other hand. They describe completely different particles, which cannot be merged). One conclusion of this manuscript is that

data on NSP in the stratosphere is limited and very heterogeneous in dependence of different variables (time, location, height etc.). This is true but this problem cannot be solved by the merge of individual published observations from different locations as presented in this manuscript.

Following the cited papers it is obvious how difficult it is to receive evidences (e.g. a link of specific source of NSP to a specific particle size or chemical species). Nevertheless, in this manuscript sometimes the "best guess" is presented as new finding.

In chapter 3,6 and 7 new data (LOAC) from balloon measurements are presented (sections 3, 6, 7). This data is presented very shortly and it is not really embedded in the review part. It is mentioned that 135 flights were carried out, but in Figure 1-7 the results of only 3 flights are shown. No details about these measurements are given (error sources, artefact discussion, data interpretation), no quantitative data at all (data tables of original data or deduced values)), nor detailed discussion in which way the data is linked to specific questions (beside qualitative speculations). I was looking forward for the data of NSP from 135 balloon flights. But there is no quantitative new data presented in the manuscript. If the manuscript should be more than a pure literature review, more details and discussion to the LOAC data has to be presented and this part must be harmonized with the review part.

---

## Author Comment (AC1) · 16 Apr 2020

**Answers to Reviewer 1**

We want to thank the reviewer for all these useful comments that have helped us to significantly improve the paper.

**General Comments**

*Comment: Starting as a nice overview of the literature on stratospheric absorbing aerosol, this paper becomes much less convincing from the moment that experimental results are analysed, and mainly interpreted. The authors often provide personal interpretations as if they were evidences, without the necessary care. Their references to the literature is also sometimes suffering from a biased or incomplete report on what is written in the cited papers. The authors lump together carbonaceous aerosols, black carbon, meteoritic smoke particles, and particle containing non-volatile residues. Further (Section 5.3), they put the emphasis on unusual processes without any reference to their real impact or importance. They give a perception of "huge amounts of absorbing particles" without real distinction amongst them, while there are large differences in optical and microphysical characteristics e.g. between a sulfate aerosol droplet that grew on a non-volatile condensation nucleus (possibly of meteoretic), soot particles from anthropogenic origin, or meteoritic smoke particles. The general view they are giving on stratospheric aerosols is sometimes very chaotic, and in some cases, biased and not in accordance with the literature they are citing. Nowhere, the authors give any perspective about the exact importance of all kinds of absorbing particles with respect to the entire stratospheric aerosol population.*

Answer: We thank the reviewer for the time he spent to review our manuscript. Perhaps, due to the length of our manuscript and the complexity of the topic, it was not easy to understand that our methodology consisted in lumping together the various nature of particles. As shown by the in situ collecting instrument, a large variety of particles can be simultaneously present during the collection and this is the only technique to properly distinguish between absorbing particles. However, the collection technique has been (and can be) operated within the stratosphere only on very rare occasions and as a result cannot be used to give any perspective about the relative and absolute importance of all types of absorbing particles. It is not possible to provide a clear absorbing-particle type distinction with our own in situ optical particle counting observations using regular balloon soundings. From the analysis of these specific observations, the best we can do is to focus on indications about the general signature of absorbing particles by investigating 1) concentrations, size distributions and speciation indices in non-ubiquitous (or transient) stratospheric layers presenting enhanced-concentrations and 2) the possible link with known seasonal meteor shower episodes.

Without providing the cases, we do not understand what the reviewer means by "in some cases, biased and not in accordance with the literature they are citing". Because of their variabilities, it is not easy to estimate the importance of NSP, because of their variability, with respect to the entire stratospheric content from all the already published sparse measurements.

*Comment: Furthermore, the style of this paper alternates between a review paper (sections 2, 4, 5) and the presentation of personal work around LOAC (sections 3, 6, 7), for which the authors do actually not provide any serious analysis nor any solid, rigourous, and convincing results. LOAC results presented here have a particularly poor information content.*

Answer: We would have needed more details to address the reviewer's comment (which points exactly need to be improved). However, we have tried now to provide a more solid analysis of the LOAC results. Please note that we have added in the analysis the flight performed until the end of 2019 (instead of mid-2019 during the previous version of the paper).

*Comment:* The article includes a lot of speculative interpretations without any attempt to validate them using some quantitative data, what is particularly pernicious: the risk is high that conjectures without any solid ground are further cited without too much care (as the authors sometimes do here) and become supposed pieces of truth after a few citations.

Answer: Although here again we would have needed point-by-point details to address the comment we understand the reviewer's concern. However, we do not include "a lot of speculative" interpretations. Our goal is to provide the contribution on aerosol understanding of the new and regular in situ observations by the LOAC instrument in the large context of references mentioned in the first part of the manuscript and finally, gathering the overall information from the new LOAC datasets and reported, to point out the numerous open questions that are still remaining on this topic, which are very different than pure conjectures.

*Comment:* The authors should much more valorise this review effort by starting from the categories identifies in Table 1, and to analyse the different aspects (particle size, chemical composition, etc.) for each identified aerosol type. They should clearly separate this review and the LOAC results, and if they choose to use these results, they should propose a solid, grounded, in-depth and quantitative analysis. Anyway, this paper needs a lot more rigour, e.g. through appropriate quantifications of their statements.

Answer: The previous associate editor has asked us to split the paper in two parts, one for the review and one for the LOAC results. The new associated editor has asked us to merge the two parts. Now it seems that the reviewer suggests to go back to the first state. It seems very difficult to reconcile all these points of view. From that point, we do not see any strong interest in rearranging again the structure of the manuscript (i.e. here basing it on the list in Table 1).

The review of various reported works show that all these studies are very difficult to unify because they are sparse (in space and time) and obtained with very different techniques with their own specificities and limitations. This was the goal of the first part of the manuscript. Then, the new in situ observations by LOAC are presented as a support to the investigations about the origin and space-time distributions of non-sulfate particles. The results are then matched to reported studies with the techniques described in the first part of the manuscript. Once again, for LOAC measurements we do not pretend to be able to comprehensively derive the physical and chemical properties of each type of absorbing particles as mentioned in Table 1.

Nevertheless, we have tried to address all the reviewer comments.

**Specific comments**

*Comment:* L. 61: I suggest slightly modifying the formulation to take into account the influence of the volcanic eruptions mentioned in the preceding sentences, since such eruption (e.g. Nabro in 2011) can really bring temporarily the atmosphere out of this background situation. For instance, the authors could add "out of such volcanic events".

*Answer:* Correction done

*Comment:* L. 73: Do the authors mean "pure solid or liquid particles"?

Answer: We have changed to: ",...can be in solid form only, but also can be externally-mixed (i.e. simultaneous presence of pure solid particles and pure liquid particles)...".

*Comment:* L. 105-108: This sentence is very similar to the sentence on L. 26-29, and in some extend, to the one on L. 1016-1018. The authors should avoid such redundancy. They can explain, for instance, that this section proposes an overview of studies on NSPs detection using all kind of platforms listed in the previous sentence.

Answer: Lines 26-29 are in the abstract, and lines 1016-1018 are in the conclusion. Thus, it is not redundancies. We have added: "This section proposes an overview such studies on NSPs using all kinds of platforms."

*Comment: L. 133: "(: : :) which backscatter light is well-known": what do the authors mean?*
Answer: We have changes to: "…which backscattered light signature …"

*L. 151-152: This sentence might requires citations.*
Answer: References are already given in the line before.

*L. 160-161: This sentence requires also a citation. I guess the sensitivity to the complex refractive index depends on the measurement principle used in in the particle counter. Maybe the text should be qualified in this sense.*
Answer: We have added the reference "Renard et al., 2005; Eidhammer et al., 2008".

*L. 162-164: This is actually a very general issue in aerosol retrieval from optical measurements, also by remote sensing.*
Answer: We agree it is a very general issue. But we think it is important to write it.

*L. 167-169: "The fraction (: : :) not composed entirely of (: : :) "is a strange formulation. I guess there are other possibilities than water and sulphuric acid to form volatile materials. The authors should thus qualify their sentence (e.g. "i.e. mainly water and sulphuric acid"). "The difference between both channels": difference of what? Please be more specific.*
Answer: The sentences are rewritten:" A more complete methodology consists in evaluating the total aerosol concentration and that of non-volatile materials, from two different collecting channel (the latter being heated to 250°C). The fraction of stratospheric particles, which are not entirely composed of volatile material (mainly water and sulfuric acid), is then evaluated by calculating the difference between these two channels (Curtius et al., 2005)."

*L. 172: "is used": I guess the list of measured parameters depend on the study and on the instrument. "Can be" might be more exact.*
Answer: correction done.

*Section 2.5: Although it is suggested in L. 225-226 through the mention of the assumption on the refractive index, it would be worth to explicitly mention that satellite measurements in the UV-visible spectral range cannot provide information on the aerosol composition (contrarily to satellite measurements in the IR such as MIPAS).*
Answer: This is not true. If measurements are conducted in the whole UV-visible domain (as done by GOMOS/Envisat), the spectral variation can provide some indication on the nature of the aerosols (from the color effect or large-band absorption lines, see e.g. Salazar et al. (2013)).

*L. 263-266: What do the authors mean? The use of a priori hypotheses on shape, composition and size distribution is necessary. Do the authors mean: "the impact of the use of a priori hypotheses"? Concerning the citation of Bourassa et al., 2012, the authors might usefully prefer Rieger et al., Stratospheric aerosol particle size information in Odin-OSIRIS limb scatter spectra, Atmos. Meas. Techn., 7, 507-522, 2014 where a discussion of the retrieval sensitivity to such parameters is discussed.*
Answer: Sorry, a part of the sentence was unclear. We have changed it to "…to follow its evolution. The a priori …" and we have added the Rieger et al. reference.

*L. 268: For the completeness, I suggest that the authors also cite Kovilakam and Deshler,*

*On the accuracy of stratospheric aerosol extinction derived from in situ size distribution measurements and surface area density derived from remote SAGE II and HALOE extinctions measurements, J. Geophys. Res. Atmos., 120, doi:10.002/2015JD023303, 2015. This paper presents an important correction brought to the processing of the optical particle counting measurements by the team of the University of Wyoming.*

> Answer: We agree, and we have added the reference.

*L. 322-323: The effect of porosity effects is less taken into account in size retrieval studies, and a citation should be required.*

> *Answer:* We agree with the reviewer. The meaning of "diameter" in case of highly porous particles is not obvious. Different definitions can be used (projected surface, gyration radius, electric mobility diameter, aerodynamical diameter…), but it is not the scope of this paper to discuss the significance of each diameter definition and the effects on comparisons between different instruments. The aerosols counters do not consider this "diameter problem" in presence of porous particles. We have added the Munoz et al. 2001 reference concerning the dependence of the scattering properties of solid aerosols depending on their shape.

*L. 328-332: The number and range of provided size classes (19 sizes classes in the 0.2-50 mm) is very large, and it might be useful to give some more insight on the uncertainty as a function of the size range. In addition, I recommend providing a reference where the uncertainty aspects of LOAC measurements are discussed.*

> Answer: The uncertainty in LOAC diameters were presented in a previous paper; nevertheless, we have added these values once again. It must be notice that the others aerosol counters do not provide errors bars on their size determination. We have added: "The uncertainties in the size determination is of ± 0.025 μm for particles smaller than 0.6 μm, 5% for particles in the 0.7-2 μm range, and of 10% for particles greater than 2 μm." The reference for the uncertainties on LOAC measurements are already given (Renard et al., 2016a, 2018).

*L. 344-354: This requires a citation.*

> Answer: The citation is already given (Renard et al., 2016a).

*L. 369-371: Do the authors mean that a different kind of balloon is used at Aire sur l'Adour, or is it some kind of nomenclature used at this place or by CNES? Please clarify.*

> Answer: We have reorganized the paragraph and modified the sentence: "in this case the balloons are called "Light Expandable Balloons" since they carry a scientific instrument that can be recovered (in case of landing on the ground), to distinguish them from conventional weather balloons. The duration of the flight is of about 2 hours."

*L. 375: The profiles in Figure 2 do not show monotonic decreasing profiles with altitude. Please be specific.*

> Answer:  Individual profiles in the stratosphere like the one shown in Figure 2 (now figure 3) are not systematically monotonically decreasing and this kind of vertical dependence with here smaller aerosol concentrations at the tropopause (here near 12 km) than a few km above has already been reported for some observations by the STAC aerosol counter designed at the LMD laboratory (see Renard et al., 2005; 2008; 2010). This can be due to vertical shear in the wind profile (thus an air mass history investigation must be conducted for all the altitude levels). However, on average, the vertical dependence of sulfate particle concentrations gives a decreasing behavior with increasing altitude has can be seen in new Figure 3 and model calculations (see SPARC 2006 aerosol assessment). We have changed the sentence to: "Globally, the aerosol concentrations decrease with altitude from the ground to the middle stratosphere. More precisely, after a small enhancement in the lower stratosphere

(Junge layer), the aerosol concentrations decrease with altitude above an altitude of 20 km, as expected for sulfate aerosols. "

*Figure 2: Are the points around a concentration range of 1.E-7-1.E-5 cm-3 significant, or are they under the detection limit of the instrument? In the first case, please justify with references to a paper describing the specifications of the instrument. In the latter case, I suggest to remove them from the plot and to adjust the scale for the sake of clarity.*

> Answer: Yes of course they are significant. LOAC can detect just one particle in a size class if the particles are very luminous (typically bigger than a few micrometers). Remove this (real) measurements means removing the possibility of detecting interplanetary dust grains. Once again, the LOAC performances are presented in the previous LOAC paper (Renard et al., 2016, 2018). We have added: "…(such big particles are so luminous when crossing the LOAC laser beam that they can be easily detected even if their number concentration is very low)."

*Caption of Figure 3: In my opinion, the authors should give a more detailed description of the figure, and more particularly of the meaning of the colored regions.*

> Answer: The LOAC typology principle was already published in several papers and is largely described in section 3.2. Nevertheless, we have added: ". In figure 5, the retrieved typologies in the stratosphere (small diamonds, compared to the "typology zones" drawn from the speciation index data base obtained in laboratory) indicate …".

*L. 380-383: This explanation seems related to Figure 2. It should thus come before the mention of Figure 3. Interpretation in L. 381-383 is highly speculative and not sufficiently scientifically grounded: I guess many other reasons might explain the presence of one large particle at that place and time. The authors should remove this statement lacking of evidence.*

> Answer: We prefer to discuss first the concentration and the nature of the particles <3 μm in the vertical profile, and then to present the detection of the largest particles.
> The reviewer has misunderstood what we have written. "Since the flight was conducted while the permanent Perseids meteor shower took place, one could suggest that LOAC has detected some dust particles coming from space". We have said" one could suggest", not "we could suggest". We present later in the paper the possible sources of this kind of particles and we say at the end of the paper that almost all these particles have terrestrial origin. We have added in the sentence: "…although other terrestrial sources are possible or a contamination effect by the balloon above the instrument cannot be totally excluded."

*L. 399-403: This criterion is not clear. How do the authors measure the "background" used as reference to assess the "strength of the concentration". Are they comparing values of the size distribution at a same altitude? This needs to be specified. Do the authors mean that they try to find bimodal size distributions, or discontinuities within a size distribution?*

> Answer: Indeed, our text is confusing, since we use the word "background" for two different things. We have changed the text to: "LOAC has detected some significant vertical variability of aerosol concentrations from one flight to another, but also during a same flight, although measurements were conducted in expected volcanically-quiescent conditions. During the same flight, we consider that a strong concentration enhancement is detected when the concentrations are at least 5 times higher than concentrations above and below the enhancement, for at least 3 consecutive size classes".

*L. 409-411: Again, the interpretations of the origin of these particles are highly uncertain and the authors should just be content with the mention that, in view of the speciation index, these particles are likely to be absorbing, in agreement with Figure*

*5. Examples of possible composition inferred from the database mentioned in L. 345
(e.g. carbonaceous particles) might be given, but with appropriate references to other
works to support their conclusions. As shown here, only the character "transparent",
"semi-absorbing" or absorbing" may be inferred.*

Answer: In section 3.3 we aim at reporting LOAC in situ observations pointing out aerosol vertical profiles and associated typologies deviating from the vertical and optical signal expected for pure sulfuric acid particles in background stratospheric conditions (i.e. free from any volcanic and fire influence). Given the structure of the manuscript, this kind of behavior and associated references are provided and discussed in chapter 4 as a matter of interpretation. Also, we did not mention that satellite data (e.g. OMPS) do not show any aerosol extinction enhancement related any specific event (e.g. fire plume) in summer 2016.

As a result, we have removed the statement about "carbonaceous particles" and have changed the text to: " "The question is whether these detected particles are linked to the Perseids period (with the fragmentation of a larger fluffy particle from space) or have a terrestrial origin.

*L. 410-411: It is hard to distinguish something special between 8 and 12 km: For
some particle classes (_ < 0.5 _m) the vertical profile is increasing toward lower
altitudes, for other (_0.5-3 _m), it present a peak at 8 km, and for larger parti-
cles (_3-50 _m), it decreases toward lower altitudes with typical concentrations of
_1.E-3-1.E-4 cm-3. Furthermore, these results are very different from results of
cirrus size distribution at mid-latitude presented by Zhao et al, J. Am. Met. Soc.
https://doi.org.10.1175/2010JAS3354.1, finding typical size distributions with a maximum
concentration around 5 to 30 _m from airborne in situ measurements and remote
ground-based measurements, while the size distributions found here between 8 and 12
km present a minimum around 10-40 _m. Hence, the authors should present convincing
arguments to interpret these particles as cirrus clouds, or remove this sentence.*

Answer: The reviewer seems to have made a mistake. First, the humidity sensors onboard the gondola shows a humidity increase at these altitudes. Secondly, the size distribution shows a concentration enhancement in the 8-30 μm size range (see below), not a minimum as said by the reviewer, thus in agreement with the Zhao et al work. Finally, the LOAC speciation index exhibits strong oscillation with size, typical of the presence of ice particles. Thus, LOAC has really detected some cirrus. The aim of our paper is not to discuss results concerning the cirrus detection. We have just mentioned their presence.

[Figure]

[Figure]

*Figure 7: It is strange that, contrarily to Figure 5 where two clearly different groups
of points, spread in different "speciation zones", suggest at first sight the present of
two different modes, in Figure 7, all indicated points seem to belong to a same group,
spread over the 3 provided zones and even out of them. This seems to suggest,*

*either that LOAC measurements cannot analyse adequately such case, or that another particle type might not be represented on the plot. Also strange is the fact that the yellow zone is obviously different in Figure 5 and 7, while the red and blue zones look similar. What is the reason for this discrepancy?*

Answer: Indeed, an error occurs when plotting the speciation zones. This is now corrected.

We have changed the text to: "This time, the typology indications (Figure 10) cannot be easily interpreted; one explanation could be the presence of a mixture of liquid and solid optically absorbing particles, having a composition evolving with size."

*L. 437-447: "As a result" of what? I really don't know from which element the authors can conclude that LOAC and STAC measurements detected similar concentration enhancements: so far, no single STAC measurement was presented! Concerning the "strong gradients" at the considered altitudes in Figures 4 and 6, there is indeed a sharp structure showing a clear change in the air masses. However, it seems that all vertical profiles roughly undergo a jump of a similar amplitude, possibly suggesting that mainly the total particle number density is changing, but that the shape of the particle size distribution is hardly affected. This would undermine the conclusion drawn in the present discussion. Therefore, it is very important that the authors provide such an illustration of the particle size density as a function of the particle size, for the considered altitude and for the closest altitude levels (say, 3 or 5 altitude levels around 18 km for Figure 4, and around 28 km for Figure 6). In this way, they could verify that the size distribution is clearly different at the level presenting a strong gradient. Also, if the authors want to make some statistics about "strong enhancements" (L. 439-440), they should provide some quantitative criteria to define such observation. They should also provide uncertainty estimates to correctly assess the strength of the phenomenon. Overall, the conclusion that most of the concentration enhancements corresponds to a dominating population of NSPs is clearly premature.*

Answer: This part is totally rewritten. We have moved the STAC measurements discussion in part 4.2. We have added two new figures on the size distribution. We have changed the text to: "Figure 5 presents an example of a strong concentration enhancement in the lower stratosphere at an altitude of 18 km, as observed during a flight conducted from Aire sur l'Adour on 11 August 2016, during the Perseids period (note that the large particles between 8 and 12 km altitude correspond to a cirrus cloud). The size distribution at 18 km differs from those below and above the enhancement, with several particles bigger than 5 μm detected at this altitude (Figure 6). The typology (Figure 7) indicates that particles up to 2 μm are indeed in liquid phase, in this case certainly sulfates, whereas biggest ones are classified as strongly optically absorbing NSPs. The question is whether these optically absorbing particles are linked to the Perseid period (with the fragmentation of a larger fluffy particle from space) or have a terrestrial origin.

Figure 8 presents another example of concentration enhancement observed in the middle stratosphere at an altitude of 28-29 km during a flight on 23 November 2017 from Aire sur l'Adour. This time, the enhancement deals only with submicronic particles (Figure 9). The typology (Figure 10) indication cannot be easily interpreted; one explanation could be the presence of a mixture of liquid and solid optically absorbing particles, having a composition evolving with size.

These two examples illustrate the two categories of enhancements detected by LOAC. The first category reflects enhanced concentrations for all size classes and the presence of large particles, and the second category concerns submicronic particle enhancements only without large particles. The LOAC typology measurements indicate that most of the enhancements are dominated by NSPs particles (more than 60%). Nevertheless, the speciation index varies from one flight to another from semi-transparent to strongly optically-absorbing particles suggesting that different families of NSPs or mixtures of different particle types could have been detected.

To tentatively attribute the origin of these different nature of enhancements and to evaluate the complex content of NSPs that LOAC could have detected, it is necessary to review the already published results on NSPs detections obtained by the various instrumental techniques."

*L. 459-462: If many research works (including the ones cites by the authors) show the importance of meteoritic smoke particles, this statement is much too simplistic in a paper on "the complex origin" of NSP. Also, although the mentioned authors indeed study the role and importance of meteoritic smoke particles in the atmosphere (often in a much broader scope than this particular region around 40 km altitude), I did not find in these papers any statement that extinction enhancements at these altitudes are definitely due to meteoritic smoke particles. The authors should thus be much more cautious in their mentions of the cited papers, and they should qualify their statement (e.g. "extinction enhancements are likely associated", or "the dominating role of meteoritic smoke particles has been shown at these altitude" : : :).*

> Answer: We have changed the sentence to : "The extinction excess is likely associated...".
> Also, we have added at the beginning of part 4.1: "Although sulfate aerosols are the main component of aerosols in the lower stratosphere, several authors have suggested that NSP could be present in the whole stratosphere."

*L. 463-465: Detection of absorbing aerosol requires techniques able to sound the complex character of the index of refraction, through polarization effects or scattering properties. This is the case of instruments such as MISR, OMI or PARASOL, but GOMOS does not offer such kind of possibilities. Hence, if it is possible from GOMOS to detect extinction with a spectral behaviour unlikely to be due to sulphate aerosols, it is not possible to attribute unambiguously such signature to the presence of absorbing aerosols particles. Please qualify this statement.*

> Answer: We speak of non-fully optically transparent aerosols when considering the GOMOS measurements. But if these particles are not transparent, there are optically absorbing. We have changed the sentence to: "The ubiquitous presence of non-fully optically transparent particles was also found in the extinction measurement of the GOMOS/Envisat instrument, for which the wavelength dependence of the extinctions strongly differed from the one expected for pure liquid aerosols only".

*L. 482-484: Although Baumgarder et al. (2004) indeed mentions this result, this statement is biased in absence of the complete information (e.g. Figure 4 in Baumgarder et al (2004)) because it gives wrongly the impression that absorbing particles are fully dominating the particle size distribution. This is not true and for particles larger than 0.3 _m, the non-light absorbing particle are largely dominating everywhere. The authors should provide a complete, non-biased information and specify how the whole size distribution behaves.*

> Answer: We have changed the sentences to: "On the other hand, using light scattering and incandescence measurement techniques, Baumgardner et al. (2004) studied the concentration of light-absorbing particles (attributed to BC and particles with metals) in the 0.2-0.8 µm size range above the tropopause in the northern polar vortex. They found more than 10 particles cm$^{-3}$ in that size range, with higher contents than for extra-vortex air. The light absorbing particles with size in the 0.2-0.3 µm size range are more abundant by a factor of 10 than the non-light absorbing particles; conversely, non-light absorbing particles larger than 0.3 µm are more abundant up to a factor of 10 than light absorbing particles."

L. 485-487: "Similarly": What is similar? Please reword.

> *Answer: We have removed "similarly".*

*Section 4.2: In this section, the authors are putting all together all kinds of events from fires to meteoritic smoke and even a large-size meteor, all of them seemly assembled under a common characteristic of "spectacular event". I am not sure there is a real scientific interest to consider such a category "sporadic strong enhancements" without much common characteristics of origin, composition, geolocation, or any other common feature that might interest the reader.*

Answer: We understand the reviewer's concern. The aim of this part was to show that the strong enhancements detected by STAC and LOAC were detected by other instruments and how these results match with reported studies, although the origin of the enhancement could be different. We have added in the text: "LOAC has detected concentration enhancements that are mainly composed of NSPs. It is then necessary to investigate whether such kind of enhancements are confirmed by other instruments and previous works. Several authors have already reported local enhancements of NSP concentrations in the stratosphere. These sporadic features are highly variable in terms of concentrations, residence times (i.e. from the scale of days to months), and proposed origins."

We have also added: "Such particles were also detected up to an altitude of 23 km by the European Aerosol Research Lidar Network (Baars et al., 2019)." and "Such enhancements have been mostly detected in the middle stratosphere, and as for LOAC observations, are dominated either by particles of all sizes or by the smallest particles only. About 25% of the 151 LOAC and 21 STAC flights exhibit such strong enhancements."

*Section 4.3: This section is equally a catch-all of a bit of everything. It becomes rapidly clear how the authors lump all kind of absorbing aerosols together. Starting from the case of "meteorite ablation around 80 km and airplane collected particles", the discussion of this case continues with "smoke particles", "interplanetary dust" and "meteoritic debris" and becomes "NSPs" in the next sentence. There is thus an obvious confusion between all these cases, although there is no reason at all that aerosol modes as different as smoke particles, soot particles or other NSPs (maybe sulphate aerosols that grew on some meteoritic condensation nuclei) have similar general size characteristics, since their origin, chemical composition and morphology are completely different. Overall, this section looks like a messy catalogue of numbers of which I am not sure about the benefit.*

Answer: The aim of this part was not well understood. We do not want to give a messy catalog of numbers; we want to show that the sizes of the NSPs having different origin is poorly determined. We have rewritten the beginning of this part: "The size distribution of NSPs in the stratosphere was poorly estimated and could depend on the origin of the particles. Only sparse direct measurements have been obtained mainly by in situ collectors and by optical counters. Among them, the concentration enhancements of NSPs detected by LOAC are in some cases composed of small particles only, and for other cases composed of large particles up to tens of $\mu$m, showing the variability of their size distribution.

Historically, Hunten et al. (1980), combining modeling calculations of the meteorite ablation around 80 km altitude and airplane collected particles (Brownlee, 1978), has proposed a bimodal repartition of the solid material in the middle stratosphere. The authors said that particles below 0.1 $\mu$m could come from the descending smoke particles, while the largest particles could originate from interplanetary dust and meteoritic debris. Nevertheless, particles coming from space could be a fraction of the NSPs content since particles coming from Earth must be also considered."

*L. 641: "The detection of metallic spheres": Please cite the corresponding work.*

Answer: This is from the work of Ebert et al. (2016) cited in the previous sentence. We have changed to: 'metallic material".

*Section 4.4: Is it right that the two first paragraph (L. 630-650) refer to carbonaceous aerosols of anthropogenic origin, while the third one (L. 660-675) concerns meteoritic*

*particles? And what about the last paragraph (L. 676-682)? The effort of reviewing
all these works and providing an insight into the wide range of compositions of these
aerosol types is very valuable, but it would be useful to split this overview into aerosol
types (e.g. soot from anthropogenic activities, meteoritic particles from extra-terrestrial
origin, and possibly particles from unknown origin).*

Answer: It seems difficult to identify the aerosol origins based on their composition only (the isotopic composition could be a better solution, as said at the end of the paper). We have reorganized this part, and we have added at the beginning: "A large variety of shapes and compositions of NSPs have been reported by various authors, and some interpretations on the origins of these particles have been proposed." The we have added at the end of each paragraph:" These particles could have a terrestrial origin", "Some of these particles could come from space, and the others could have a terrestrial origin", "The origin of these particles is not well established."

*L. 697-702: The interest of this isolated mention of "prograde orbits" is unclear. Further,
it is difficult to understand why, just after mentioning the difficulty to categorize aerosol
types, the authors seem to look for further sub-categories.*

Answer: we have added: "The grains coming from these kinds of comets could have a lower relative velocity to Earth orbit than grains coming from retrograde orbits, and thus could more easily survive the atmospheric entry."

*L. 714-715: This statement requires revision: although this question was widely debated
in the community, it is likely the volcanic plume of the Nabro (13_N, 41_E, June
2011) reached the upper troposphere due to the eruption and was transported afterward
to the stratosphere along pathways provided by the Asian summer monsoon.*

Answer: The papers on Nabro aérosls concerns sulfate aerosols, not NSPs.

*L. 716: What do the authors mean by "organic fuel"?*

Answer: We have removed "organic fuel burning".

*L. 722; Tropopause folds are phenomena occurring in the extra-tropics.*

Answer: We have removed "at tropical latitudes."

*L. 720: Which kind of overshooting convective systems do the authors have in mind?
The three provided reference correspond to pyroconvection.*

Answer: We deal with convection likely to be enhanced by fire activity. We have added the Jost et al. (2004) reference that mentions the possibility of overshooting convective system.
We have changed the sentence accordingly: "These particles can reach the lower stratosphere through direct injection by cross-tropopause pyroconvection events or through transport associated with overshooting convection enhanced by fire activity (Jost et al., 2004; Damoah et al., 2006; Fromm et al., 2005; de Laat et al., 2012)."

*L. 725: "largely determine the composition of the UTLS": is it true that the AMA is the
determining factor of the composition of UTLS? It might be necessary to quality this
statement in "plays a major role in the composition (: : :)". This statement would require
a reference in any case.*

Answer: All the reference given in the following sentences can support this affirmation. Nevertheless, we have changed the sentence to "… (AMA) can largely…"

*L. 734-735: This sentence requires some reference. In particular, the authors should
mention important projects specifically dedicated to the study of the ATAL and its
composition such as STRATOCLIM (Brunamonti et al, Balloon-borne measurements
of temperature, water vapor, ozone and aerosol backscatter on the southern slopes*

*of the Himalayas during StratoClim 2016-2017, Atm. Chem. Phys., 18, 15937-*
*15957, 2018) or BATAL (Vernier et al., BATAL, the balloon measurements campaign*
*of the Asian tropopause aerosol layer, B. Am. Meteor. Soc., May 2018, 955-973,*
*http://doi.org/10.1175/BAMS-D-17-0014.1, 2018).*

Answer: The Vernier et al. 2018 reference was given a few lines above. We have added the Brunamonti et al. reference.

*L. 750-756: How significant are such source in the stratosphere?*

Answer: It is not known if this source is significant or not. We mentioned it since it has been published. We have added "this highly speculative process".

*L. 757-768: What is the frequency of re-entry of satellites or rocket disintegration? The probability of observing such event looks particularly low to me, and the interpretation of Ebert's results in the light of Hamdan's work looks highly improbable and speculative to me*

Answer: Re-entry satellite or rocket disintegration could be only a (small) part of the phenomena that could produce such particles. There is about one hundred rocket launches per year, and a similar number of re-entries of 'heavy' pieces of rocket or satellite. Ebert et al. (2016) has mentioned the possibility that some of the particles they have detected could come from rockets. We have added: "(more than one hundred per year)".

The similarity of the composition (Fe-NI-Cr) of some particles detected by Ebert et al. (2016) and these carbon polymeric nanocomposites could be fortuitous or not. We have changed the text to: "…it could be just a coincidence, or some of these particles could be indeed nanocomposites…".

*Section 5.3: It is a little bit strange that the overview of the NSP production mechanisms describes as second mechanism the formation of "hypothetical long-lived volcanic soot particles" and as third one "rockets exhaust and the disintegration of satellites" of which the relative importance is particularly uncertain in the global NSP production. This one is followed by "compact particles and filaments" possibly (hypothetically?) occurring during atmospheric entries of meteorites and satellites/rocket debris" , and "rare events of the disintegration of large meteoroids, with sizes above 10 m". Finally, the very last one mentions meteoritic ablation likely to be about the most important of all the cited processes. The authors should present the different mechanisms in the order of importance and impact on the atmospheric processes, starting from the most important.*
*They also should quantify in some way their importance, in terms of concentration, spatial coverage, lifetime, occurrence rate, etc.*

Answer: The reviewer is right; our presentation of the sources inside the atmosphere was confusing. We have re-organized this part following the reviewer suggestion.

*L. 795-797: Sentence is not clear.*

Answer: We have changed the sentence to: "These grains slowly spiral towards the Sun (under Poynting-Robertson effect); the persistence of the interplanetary dust cloud indicates that a more or less continuous replenishment takes place."

*L. 797: Interpretation may be arbitrary. Do the author mean "analysis"?*

Answer: We have changed "interpretation" to "analysis".

*L. 804: "faint dust bands attributed to collisions within the asteroid belt": It is not clear if the authors refer to some particle "cloud", or to some spectral band. Please clarify.*

Answer: We have changed the text to: "…attributed to originate from asteroid collisions within the asteroid belt".

*L. 808-810: Odd sentence. Please reword.*

    Answer: We have rewritten this paragraph:" Because of its sources of replenishment, the interplanetary dust cloud is not a featureless structure (e. g. Levasseur and Blamont, 1973), although the dust particles are progressively randomly distributed. When interplanetary dust particles impact the atmosphere, they induce the formation of sporadic meteors, meteor showers, or even meteor storms from more recent dust particles injected in the interplanetary cloud."

*L. 812-813: Which cloud? This sentence requires some reference. Idem for L. 813-817.*

    Answer: We give here well-established background information on the interplanetary dust clouds that do not require references.

*L. 826-828: Incorrect sentence: What is "prevalence influx"? Speed is a physical parameter, thus singular.*

    Answer: We have changed the sentence to: "The speed of interplanetary dust particles coming from JFCs and that impact the Earth's atmosphere are mostly low, around 15 km/s range (Nesvorný et al., 2010) or less."

*L. 829: "relative velocity": with respect to what?*
*L. 829-830: The authors misuse results published by Levasseur-Regourd and Lasue (2011). These authors draw their conclusions about the probable "survival" of significant amount of organics for seeding life on early Earth, during the Late Heavy Bombardment epoch, "while the spatial density of dust in the interplanetary dust clouds was orders of magnitude greater than nowadays".*

    Answer: Indeed, this sentence was out of the scope of our paper. We have removed the sentence.

*L. 841-842: The meaning of the use of italic characters is unclear.*
    Answer: We have corrected this typo error.

*Section 5.3: The authors give a lot of details about interplanetary dust particles and discuss extensively the origin of these particles (comets or asteroids). However, the conclusions of interest for the study of NSP is particularly poor and vague: enhancements of interplanetary aerosol particles [of carbonaceous aerosols] might show similar properties (which ones ?) as aerosol particles of terrestrial origin. The authors should shorten this section.*

    Answer: The goal of this section is to indicate that NSPs potentially have a link with cometary material which is often neglected. We have added: "When considering the material coming from space and found in the Earth atmosphere, some papers consider meteoritic material but not dust coming from comets. Since cometary material could differ from meteoritic material, we propose below a short review of the physical properties of the dust present in the interplanetary cloud."

*L. 866-867, Figure 9: Does it mean that STAC and LOAC flights span several weeks? This is not what previous text suggests (e.g.:, Figures 1 to 7, corresponding to flight on a single day). The authors should clarify what they mean on Figure 9, and how many flights are concerned. A table with an overview of all types of flights and their characteristics might be useful.*

    Answer: We have added in part 3.3: "The duration of the flight is of about 2 hours."
    Indeed, the Figure 9 (now 12) was unclear. We have added in the text: "To search for a possible seasonality in the occurrence of the enhancements, all the LOAC and STAC flights are binned in each week during which the measurement was performed. Then, the number of concentrations enhancements detected each week is calculated, and is divided by the number of flights performed

during the week for normalization. The results are plotted in Figure 12 as a function of the day of the year." And we have changed the Figure 12.

*L. 863-865: On which criteria do the authors base their statement? How fortuitous such event is depends on its persistence and frequency. Based on the potential total daily amount (up to 270 tons) given in L. 833, meteoritic smoke particles might be not so fortuitous.*

      Answer: The reviewer is right. It is difficult to provide an estimate of the probability. Thus we have remove the sentence.

*L. 869-872: Did the authors calculate any correlation?*
*L. 871-872: I do not see any convincing, quantitative analysis that might lead to this conclusion.*

      Answer: We have rewritten the first sentence to: "It seems that no obvious correlation can be found between the variability of the number events and the meteor shower dates and intensities, although some fortuitous coincidence can exist", and we have added: "Only 15% of the enhancements occur in a period starting 3 days before the maximum intensity of the meteor shower and ending 7 days later".

*L. 880-917: The authors list a full page of speculative explanations, without proposing any serious analysis on how their measurements may support any of them. I have some doubt about the relevancy of such approach.*

      Answer: We have presented just above and in part 3.3 a more detailed analysis of the enhancements. We have rewritten the sentence to: "Different dynamical processes may be proposed to explain the various concentration enhancements of NSPs detected by our aerosol counters and previously reported by other authors at different dates and from different locations."

*L. 918: "double repartition" is incorrect.*

      Answer: We have changed to "bimodal".

*L. 921: "This double repartition looks like the one proposed by Beresnev et al. (2017)": this is again a very weak "analysis" referring to a non peer-reviewed proceeding, with a vague description ("the [double repartition] for the accumulation layers of fractal and spherical carbonaceous particles respectively"). Such a superficial analysis is totally insufficient.*

      Answer: The work of Cheremisin et al. (JGR, 2011 doi:10.1029/2011JD015958) has revealed the possibility of the photophoretic effect to exceed gravity and to be responsible for the formation of layers of optically-absorbing particles at 20 and 30 km. We more focus here on the work of Beresnev et al. (2017) because it is more recent and deals with fractal particles more representative of soot particles and aggregates. The Beresnev et al (2017) paper is in a peer review journal, not in non peer-reviewed proceedings. We have added: "Cheremisin et al. (2011) has revealed the potential of the photophoretic effect to exceed gravity and to be responsible for the formation of layers of optically-absorbing particles at 20 and 30 km. The direction of the photophoretic force can be opposite to gravity and it can exceed the weight of the particle. In particular, particles larger than 1 $\mu$m can be suspended in the atmosphere due to this effect (Cheremisin et al., 2005)." Later, we have changed this paragraph to: "This bimodal repartition looks like the one proposed by Beresnev et al. (2017) for the accumulation layers of fractal particles that could correspond to dense aggregates (density of 2 g.cm$^{-3}$) and of fluffy aggregates (densities in the 0.16-0.35 g.cm$^{-3}$). In a steady-state atmosphere, under the action of the solar photophoresis the accumulation layers could be in the 13-23 altitude range for the dense aggregates, and in the 25-30 km altitude range for fluffy aggregates, thus close to the ones detected by LOAC and STAC (the LOAC typologies indicate that indeed the optically-absorbing particles dominate the aerosol-enhanced layers)."

*L. 925-927: What does it mean? What do the authors want to prove?*

Answer: We have changed the text to: "It seems difficult to determine the origin of the particles inside the enhancements, based only on the altitude of the layers and thus on the porosity of the particles. Carbonaceous particles coming from space, produced within the atmosphere and emitted from the Earth's surface can be compact and/or fluffy depending on their production mode and on their aging in the atmosphere."

*L. 945-946: Following Figure 11 and taking into account the error bars, there is no statistically significant decrease of the concentration with increasing altitude.*
*Caption Figure 11: "Evolution" (should be singular) seems more a time-related concept.*
*"Altitude dependence" seems more adequate. What is "mean evolution"? Please specify what the circles represent.*

Answer: Our text was confusing. The horizontal bars do not represent the errors, but the interannual variability, which is not the same. We have hanged the legend to: "Altitude dependence of the concentrations of the large particles detected by LOAC for the 2013-2019 period, for 3 size classes; the dots represent the mean values, and the horizontal bars represent the interannual variability"

We have added "tentatively" in the first sentence of this part. We have changed the text to: "Since 151 flights were performed in the mid-2013 – 2019 period, it seems statistically reasonable to calculate the mean values of particle concentrations for three broad size classes (5-7.5 $\mu$m, 10-17.5 $\mu$m, 20-50 $\mu$m) in layers of 5 km width. Figure 14 presents the altitude dependence of the mean concentrations; the horizontal bars represent the interannual variability. The concentrations decrease by about of factor two from 15 to 35 km for the three size classes."

*L. 948-949: Even if the ensemble of the considered measurements are likely to provide much useful information, it is impossible to conclude on any trend from the results given in Figure 11 (profiles and their error bars) and the fact that the total analysed volume of air represents about 1 to 2 cm3 of the whole stratosphere.*

Answer: We disagree with the reviewer comment. First, it is not a sampling in the order of a few $cm^3$ but of $m^3$ (as said in the text) so a factor of $10^6$ more. Secondly, if the results were not statistically significant, the 3 vertical profiles should oscillate and would not exhibit similar altitude dependence.

*L. 949-951: What do the authors intend to prove by this statement? What is the available measurement sample and the representativeness of these observations?*

Answer: We have removed this sentence that does not provide significant information.

*L. 952-961: This "analysis" is inadequate. How do LOAC flights match with the RECONCILE campaigns inside the polar vortex? Even by summing the contributions to all panels of Figure 11, the result does not reflect in any way a concentration of about 1000 particles/m3 for particles greater than 3 microns below 21 km. And saying that LOAC measurements agrees with cited results at 30 km and 38 km respectively, is meaningless. The authors should remove all this discussion.*

Answer: We prefer to maintain this discussion, since these measurements of large particles in the stratosphere are rare; it is of interest to compare the LOAC results with previous measurements.

We have changed to text to: "Ebert et al. (2016) have collected during the RECONCILE campaign inside the polar vortex about $10^3$ particles $m^{-3}$ greater than 3 $\mu$m in the lower stratosphere (below 21 km). At such altitude, the LOAC concentration of particles greater than 5 $\mu$m is below 150 particles $m^{-3}$; the concentration for particles greater than 3 $\mu$m increase to about 400 particles $m^{-3}$ (although some of them could be liquid), which is not so far from the Ebert et al. (2016) value.

Using the Brownlee (1978) measurements, Hunten et al. (1980) have estimated the concentration of interplanetary dust (or micrometeorites) at 30 km to be of about $10^{-3}$ particles $m^{-3}$ for sizes greater than 5 μm, On the other hand, the concentration of collected particles greater than 5 μm by the DUSTER instrument during one balloon-flight at 38 km was of about one particle $m^{-3}$ (Della Corte et al., 2013). These concentrations are well below the LOAC mean value of about 50 particles $m^{-3}$, although the annual LOAC interannual variability indicate no detection of large particles at such attitude can sometime occur.

All these sparse measurements have sampled a small volume of the stratospheric air, which can cause some variability for the estimation of large particle's concentrations. These preliminary results show that more measurements will be needed to better estimate this content."

*L. 973-975: This reference is difficult to find, and the mentioned rate of 10% is not mentioned in a summary found on internet. However, the title of the paper specifies that this work concerns micrometeorites collected at ground level in ultra-clean snow in Antarctica. This is far from representative of the interplanetary dust flux reaching the (global) stratosphere. The rest of the section (L.975-1011) is an additional hope of speculation without any attempt to test them against any quantitative result.*

Answer: We have removed Dobrica et al. reference and the part of the sentence on the particles reaching Earth surface.

We disagree with the reviewer comment. It seems necessary to discuss the possible origin of such discrepancies between the in situ measurements and the estimated concentrations of large particles coming from space. We have added: "Many more in situ measurements of large particles will be necessary to better estimate the concentrations of large particles in the stratosphere. At present, we could only propose several explanations to tentatively explain these discrepancies:".

*L. 1030-1034: The developments as presented in the paper do not allow drawing these conclusions.*

Answer: Following all the previous reviewer comments, the new elements we have given on the LOAC measurements and the results could better support our conclusions.

*L. 1072-1076: These sentences include generalities without any relevant added information. I suggest removing them.*

Answer: We agree with the reviewer. We have removed this sentence.

**Technical comments**

*L. 27: missing comma.*
*L. 35-36: "ranging between 17 and 30 km altitude"?*
*L. 227: "They cannot access the local variability" or "They give no access to the local variability".*
*L. 114: "sparse of time-series measurements": looks odd. Maybe "sparse or continuous", "sparse or routine measurements"?*
*L. 118: "abundance". Physical parameter, thus singular.*
*L. 127: "EARLINET".*
*L. 241-243: Odd sentence (combination of "often" and "in general").*
*L. 244-246: "such as volcanic eruptions, injection (: : :), or pyroconvection"?*
*L. 264: "hypotheses".*
*L. 313: "can be"?*
*L. 333: I suggest "The particle size provided by LOAC" to avoid the strange association "particles provided by LOAC".*

*L. 355: "corrected for".*
*L. 373-374: "an example of vertical (: : :)"; "on 17 August 2017"*
*L. 444, Figure 3 (and maybe elsewhere): "absorbing" instead of "absorbent".*
*L. 704: "main characteristics".*
*L. 752: missing ")".*
*L. 793: Please correct the sentence.*
*L. 795: "with' instead of "within".*
*L. 817: missing '.'*
*L. 971: "originating".*
*L. 1175: "boundary".*
*L. 1263: "Fadnavis, S.".*
*L. 1330: "Process."*
*L. 1595: "OSIRIS".*
*L. 1014: "Conclusions" is Section 8.*

Answer: We have made the corrections proposed by the reviewers.

---

## Author Comment (AC2) · 16 Apr 2020

**Answers to Reviewer 2**

We want to thank the reviewer for these useful comments that have helped us to significantly improve the paper.

*The complex origin and spatial distribution of non-pure sulfate particles (NSPs) in the stratosphere, by Jean-Baptiste Renard et al. gives an overview of the literature on stratospheric aerosol. With more than 100 cited papers a lot of diverse information about stratospheric particles is compiled. In this "review"-part (chapter 1,2,4,5) specific results from diverse papers are mixed together to present an overall picture. But in some cases this presented overall picture is inaccurate as not all presented results can be mixed in reality because of a complete different data base in the cited papers (e.g. particle composition of small carbonaceous particles presented by Ebert et al., 2016 on the one hand and Schütze et al., 2017 on the other hand. They describe completely different particles, which cannot be merged). One conclusion of this manuscript is that data on NSP in the stratosphere is limited and very heterogeneous in dependence of different variables (time, location, height etc.). This is true but this problem cannot be solved by the merge of individual published observations from different locations as presented in this manuscript.*

Answer: The reviewer is right; it was the aim of the review of previous works on NSPs to show this problem. We have added at the beginning of part 4: "Although sulfate aerosols are the main component of aerosols in the lower stratosphere, several authors have suggested that NSP could be present in the whole stratosphere. Nevertheless, an overall picture is difficult to establish because results on concentrations, size distributions and compositions stem from different works using different methodologies, and can concern different origins and compositions of particles, which cannot be merged. Thus, the results presented below show the complexity of the NSP content, which is not fully assessed by the studies already conducted."

We have also added at the beginning of part 4.4: "A large variety of shapes and compositions of NSPs have been reported by various authors, and some interpretations on the origins of these particles have been proposed."

*Following the cited papers it is obvious how difficult it is to receive evidences (e.g. a link of specific source of NSP to a specific particle size or chemical species). Nevertheless, in this manuscript sometimes the "best guess" is presented as new finding.*

Answer: It is difficult to answer. Lots a work needs however to be still done in further studies to confirm our first work. We hope that all the answers we have given to reviewer 1 will satisfy this this comment.

*In chapter 3,6 and 7 new data (LOAC) from balloon measurements are presented (sections 3, 6, 7). This data is presented very shortly and it is not really embedded in the review part. It is mentioned that 135 flights were carried out, but in Figure 1-7 the results of only 3 flights are shown.*

Answer: Since we present new measurements, these data cannot be embedded in the review part. On the other hand, indeed the LOAC data were presented too briefly.

We have added at the end of section 3.2: "LOAC can be used frequently to performed measurement in the stratosphere to evaluate the vertical and temporal variability of the aerosol content, to identify the presence of NSPs, and also better determine the concentration of large particles greater than 5 mm and up to 50 mm."

We have added in the section 3.3 a figure showing the total concentrations for all the flight and the text: " The mean vertical evolution of number concentrations of particles greater than 0.2 $\mu$m can be estimated considering all LOAC flights. At each km, the histogram of the concentrations is calculated and is fitted, in log scale, with a lognormal function to estimate the mode corresponding to

the most frequent concentration. As expected for background stratospheric aerosols, the vertical profile of this mode, in red in Figure 2, decreases with increasing altitudes. The individual profiles fluctuate on both sides of the mode profile, probably due to the local variability of the aerosol content. Nevertheless, the variability is higher for the greatest concentrations, with several strong increases of km-width. Such kind of increases reflects those previously reported by Renard et al. (2010) in the middle stratosphere using another balloon-borne aerosol counter.

Examples of individual profiles are presented below, to illustrate the variability of the stratospheric aerosol content."

[Figure]

Figure 2: Vertical evolution of the concentrations of particles greater than 0.2 μm considering all LOAC flights; the red line corresponds to the mode of the most frequent concentrations and the green lines correspond to the first and third quartiles

*No details about these measurements are given (error sources, artefact discussion, data interpretation), no quantitative data at all (data tables of original data or deduced values)), nor detailed discussion in which way the data is linked to specific questions (beside qualitative speculations).*

Answer:

Error sources:

The uncertainties measurements (and thus the errors sources) were already given in part 3.2. We have added: "The uncertainties in the size determination is of ± 0.025 μm for particles smaller than 0.6 μm, 5% for particles in the 0.7-2 μm range, and of 10% for particles greater than 2 μm."

It is not here the place to discuss in detail the artefact of aerosol counters since it is a well-known technique of measurements. We have already written at the beginning of part 3.2: "Conventional aerosols counters typically performed measurements at large scattering angles (greater than 30° and often around 90°). Since the scattered light at these angles is sensitive to the size of the particles but also to their complex refractive index and their shape including porosity effects (e.g. Muñoz et al., 2001), conventional optical counter measurements must be corrected for the nature of the particles. On the opposite, LOAC performs measurement at small scattering angles, in the 11°-16° range, where the scattered light by irregular particles is mainly coming from diffraction that does not depend on the complex refractive index nor on the porosity of the irregular-shaped particles (Lurton et al. 2014; Renard et al., 2016a)."

If the reviewer is referring to the concentrations enhancements, we have already written: "During the same flight, we consider that a strong concentration enhancement is detected when the concentrations are at least 5 times higher than concentrations above and below the enhancement, for at least 3 consecutive size classes of submicronic particles (this criterion is to ensure that the

enhancements are real and are not due to noise measurement fluctuations). We exclude the measurements conducted at the edge of the polar vortex where the local dynamical variability can affect the aerosols content (Renard et al., 2008)."

Data interpretation:

We have tried to improve the discussion on data interpretation. We provide now a figure of the size distribution for the two examples of concentration enhancements.

Indeed, we have not provided tables of the results since we prefer to present figures. Interpretation of the data are presented in Figures 12 to 14.

Concerning the concentration enhancements, we had initially written: "The LOAC typology measurements indicate that the enhancements are dominated by NSPs particles (more than 60%)." And also: "about 25% of the 151 LOAC and 21 STAC flights exhibit strong concentration enhancements". We have added: "It seems that no obvious correlation can be found between the variability of the number events and the meteor shower dates and intensities, although some fortuitous coincidence can exist. Only 15% of the enhancements occur in a period starting 3 days before the maximum intensity of the meteor shower and ending 7 days later." To link our results to specific questions, we have added: "Different dynamical processes may be proposed to explain the various concentration enhancements detected by our aerosol counters and previously reported by other authors at different dates and from different locations."

Concerning the vertical distribution of the enhancements, we had initially written (including deduced values): "A statistical analysis of the altitudes of the concentration enhancement events detected by LOAC seems to indicate a bimodal repartition, one centered at around 17 km and the second one at around 30 km added". We have added: "This bimodal repartition looks like the one proposed by Beresnev et al. (2017) for the accumulation layers of fractal particles that could correspond to dense aggregates (density of 2 g.cm$^{-3}$) and of fluffy aggregates (densities in the 0.16-0.35 g.cm$^{-3}$). In a steady-state atmosphere, under the action of the solar photophoresis the accumulation layers could be in the 13-23 altitude range for the dense aggregates, and in the 25-30 km altitude range for fluffy aggregates, thus close to the ones detected by LOAC and STAC (the LOAC typologies indicate that indeed the optically-absorbing particles dominate the aerosol-enhanced layers)."

Concerning the large particles, we have added: "Finally, LOAC can be tentatively used to estimate the background content of large particles with sizes ranging from several μm up to 50 μm, which are obviously NSPs since large pure droplets cannot exist in the stratosphere." We have also added: "Since 151 flights were performed in the mid-2013 – 2019 period, it seems statistically reasonable to calculate the mean values of particle concentrations for three broad size classes (5-7.5 μm, 10-17.5 μm, 20-100 μm) in layers of 5 km width. Figure 14 presents the altitude dependence of the mean concentrations; the horizontal bars represent the interannual variability. The concentrations decrease by about of factor two from 15 to 35 km for the three size classes. Ebert et al. (2016) have collected during the RECONCILE campaign inside the polar vortex about $10^3$ particles m$^{-3}$ greater than 3 μm in the lower stratosphere (below 21 km). At such altitude, the LOAC concentration of particles greater than 5 μm is below 150 particles m$^{-3}$; the concentration for particles greater than 3 μm increase to about 400 particles m$^{-3}$ (although some of them could be liquid), which is not so far from the Ebert et al. (2016) value.

Using the Brownlee (1978) measurements, Hunten et al. (1980) have have estimated the concentration of interplanetary dust (or micrometeorites) at 30 km to be of about $10^{-3}$ particles m$^{-3}$ for sizes greater than 5 μm, On the other hand, the concentration of collected particles greater than 5 μm by the DUSTER instrument during one balloon-flight at 38 km was of about one particle m$^{-3}$ (Della Corte et al., 2013). These concentrations are well below the LOAC mean value of about 50 particles.m$^{-3}$, although the annual LOAC interannual variability does not indicate any of large particles at such altitude can sometime occur."

*I was looking forward for the data of NSP from 135 balloon flights. But there is no quantitative new data presented in the manuscript. If the manuscript should be more than a pure*
*literature review, more details and discussion to the LOAC data has to be presented*
*and this part must be harmonized with the review part.*

Answer: We have tried to significantly improve the presentation and the discussion of the LOAC data, and we give now more quantitative results. The presence of the accumulation layers, and their altitude dependence, the absence of correlation between concentrations enhancements and meteors showers, and the detection of large particles by an aerosol counter and their concentrations are new data. We have also indicated that almost all these particles have more probably a terrestrial origin. We have added at the end of the introduction: "As studying NSPs in the stratosphere is a complex topic, an extensive review of the state-of-the-art techniques is needed. This paper however does not limit to a literature review on this topic."